# Renal medullary carcinomas depend upon *SMARCB1* loss and are sensitive to proteasome inhibition

Andrew L Hong[1,2,3], Yuen-Yi Tseng[3], Jeremiah A Wala[3], Won-Jun Kim[2], Bryan D Kynnap[2], Mihir B Doshi[3], Guillaume Kugener[3], Gabriel J Sandoval[2,3], Thomas P Howard[2], Ji Li[2], Xiaoping Yang[3], Michelle Tillgren[2], Mahmhoud Ghandi[3], Abeer Sayeed[3], Rebecca Deasy[3], Abigail Ward[1,2], Brian McSteen[4], Katherine M Labella[2], Paula Keskula[3], Adam Tracy[3], Cora Connor[5], Catherine M Clinton[1,2], Alanna J Church[1], Brian D Crompton[1,2,3], Katherine A Janeway[1,2], Barbara Van Hare[4], David Sandak[4], Ole Gjoerup[2], Pratiti Bandopadhayay[1,2,3], Paul A Clemons[3], Stuart L Schreiber[3], David E Root[3], Prafulla C Gokhale[2], Susan N Chi[1,2], Elizabeth A Mullen[1,2], Charles WM Roberts[6], Cigall Kadoch[2,3], Rameen Beroukhim[2,3,7], Keith L Ligon[2,3,7], Jesse S Boehm[3], William C Hahn[2,3,7]*

[1]Boston Children's Hospital, Boston, United States; [2]Dana-Farber Cancer Institute, Boston, United States; [3]Broad Institute of Harvard and MIT, Cambridge, United States; [4]Rare Cancer Research Foundation, Durham, United States; [5]RMC Support, North Charleston, United States; [6]St. Jude Children's Research Hospital, Memphis, United States; [7]Brigham and Women's Hospital, Boston, United States

*For correspondence:
william_hahn@dfci.harvard.edu

Competing interest: See
page 25

Reviewing editor: Ross L
Levine, Memorial Sloan Kettering
Cancer Center, United States

**Abstract** Renal medullary carcinoma (RMC) is a rare and deadly kidney cancer in patients of African descent with sickle cell trait. We have developed faithful patient-derived RMC models and using whole-genome sequencing, we identified loss-of-function intronic fusion events in one *SMARCB1* allele with concurrent loss of the other allele. Biochemical and functional characterization of these models revealed that RMC requires the loss of *SMARCB1* for survival. Through integration of RNAi and CRISPR-Cas9 loss-of-function genetic screens and a small-molecule screen, we found that the ubiquitin-proteasome system (UPS) was essential in RMC. Inhibition of the UPS caused a G2/M arrest due to constitutive accumulation of cyclin B1. These observations extend across cancers that harbor *SMARCB1* loss, which also require expression of the E2 ubiquitin-conjugating enzyme, *UBE2C*. Our studies identify a synthetic lethal relationship between *SMARCB1*-deficient cancers and reliance on the UPS which provides the foundation for a mechanism-informed clinical trial with proteasome inhibitors.
DOI: https://doi.org/10.7554/eLife.44161.001

## Introduction

Renal medullary carcinoma (RMC) was first identified in 1995 and is described as the seventh nephropathy of sickle cell disease (*Davis et al., 1995*). RMC is a rare cancer that occurs primarily in patients of African descent that carry sickle cell trait and presents during adolescence with symptoms of abdominal pain, hematuria, weight loss and widely metastatic disease. Due to the aggressive behavior of this disease and the small numbers of patients, no standard of care exists. Patients are generally treated with multimodal therapies including nephrectomy, chemotherapy and radiation therapy. Despite this aggressive regimen, the mean overall survival rate is only 6–8 months (*Alvarez et al., 2015*; *Beckermann et al., 2017*; *Ezekian et al., 2017*; *Iacovelli et al., 2015*).

**eLife digest** Renal medullary carcinoma (RMC for short) is a rare type of kidney cancer that affects teenagers and young adults. These patients are usually of African descent and carry one of the two genetic changes that cause sickle cell anemia. RMC is an aggressive disease without effective treatments and patients survive, on average, for only six to eight months after their diagnosis.

Recent genetic studies found that most RMC cells have mutations that prevent them from producing a protein called SMARCB1. SMARCB1 normally acts as a so-called tumor suppressor, preventing cells from becoming cancerous. However, it was not clear whether RMCs always have to lose SMARCB1 if they are to survive and grow.

Often, diseases are studied using laboratory-grown cells and tissues that have certain features of the disease. No such models had been created for RMC, which has slowed efforts to understand how the disease develops and find new treatments for it. Hong et al. therefore worked with patients to develop new lines of cells that can be used to study RMC in the laboratory. These RMC cells started dying when they were given copies of the SMARCB1 gene, which supports the theory that RMCs have to lose SMARCB1 in order to grow.

Hong et al. then used a set of genetic reagents that can suppress or delete genes that are targeted by drugs, and followed this by testing a range of drugs on the RMC cells. Drugs and genetic reagents that reduced the activity of the proteasome – the structure inside cells that gets rid of old or unwanted proteins – caused the RMC cells to die. These proteasome inhibitor drugs also killed other kinds of cancer cells with SMARCB1 mutations.

Proteasome inhibitors are already used to treat different types of cancer. Potentially, a clinical trial could be run to see if they will treat patients whose cancers lack SMARCB1. Further work is also needed to determine the exact link between SMARCB1 and the proteasome.

DOI: https://doi.org/10.7554/eLife.44161.002

Recent studies have implicated loss of SMARCB1 in RMC (*Calderaro et al., 2016*; *Carlo et al., 2017*; *Cheng et al., 2008*). SMARCB1 is a tumor suppressor that when conditionally inactivated in mice leads to rapid onset of lymphomas or brain tumors (*Han et al., 2016*; *Roberts et al., 2002*). Furthermore, SMARCB1 is a core member of the SWI/SNF complex where alterations of one or more members have been identified in up to 20% of all cancers (*Helming et al., 2014*; *Kadoch et al., 2013*) including malignant rhabdoid tumors (MRTs) and atypical teratoid rhabdoid tumors (ATRTs). MRTs and ATRTs harbor few somatic genetic alterations other than biallelic loss of *SMARCB1* and occur in young children (*Chun et al., 2016*; *Gröbner et al., 2018*; *Lee et al., 2012*; *Ma et al., 2018*; *Torchia et al., 2016*). In contrast, RMC patients present as adolescents/young adults, are primarily of African descent and have been found to have fusion events in *SMARCB1* and gene mutations in *ERG*, *PDGFRB, MTOR, and ERBB2* (*Calderaro et al., 2016*; *Carlo et al., 2017*). Other pathways implicated in this disease included loss of *TP53* and *VEGF/HIF1A* (*Swartz et al., 2002*).

An unresolved question is whether these cancers depend upon loss of SMARCB1. Furthermore, there is an unmet need to identify therapeutic targets to provide better treatments for these patients. Here, we have developed and characterized faithful cell lines of this rare cancer. We demonstrate that RMC depends on loss of SMARCB1 for survival and, through integrated genetic and pharmacologic studies, we uncover the proteasome as a core druggable vulnerability in RMC and other SMARCB1-deficient cancers.

## Results

### Derivation and genomic characterization of RMC models

From September 2013 until September 2018, three patients who had a diagnosis of renal medullary carcinoma (RMC) were consented to IRB approved protocols (Materials and methods). All patients were of African descent and adolescents. We first attempted to create a patient-derived xenograft from each patient by implanting tissue in the sub-renal capsule or subcutaneously in

immunodeficient mice but these samples did not form tumors after 6 months of monitoring. We then attempted to develop cell lines from these patients and generated cell lines from two of the three patients (CLF_PEDS0005 and CLF_PEDS9001) (Materials and methods). For the first patient (CLF_PEDS0005), we obtained the primary tissue from our local institution at the time of the initial nephrectomy. We generated a short-term culture normal kidney cell line, CLF_PEDS0005_N, and a tumor cell line, CLF_PEDS0005_T1 (*Figure 1—figure supplement 1a*). In addition, we obtained fluid from a thoracentesis performed when the patient relapsed 8 months into therapy. We isolated two cell lines that grew either as an adherent monolayer, CLF_PEDS0005_T2A, or in suspension, CLF_PEDS0005_T2B. Each of these tumor cell lines expressed the epithelial marker, CAM5.2, and lacked expression of SMARCB1 similar to that observed in the primary tumor (*Figure 1—figure supplement 1b*). For the second patient (CLF_PEDS9001), we partnered with the Rare Cancer Research Foundation and obtained samples through a direct-to-patient portal (www.pattern.org). The primary tumor tissue from the second patient was obtained at the time of the initial nephrectomy. From this sample, we generated the tumor cell line, CLF_PEDS9001_T1. Cell lines were generated from patients who received 4–8 weeks of neoadjuvant chemotherapy prior to their nephrectomy.

Sequencing and cytogenetic efforts have identified deletion of one allele of *SMARCB1* along with fusion events in the second allele of *SMARCB1* in RMC patients (*Calderaro et al., 2016*; *Carlo et al., 2017*). We performed WES (CLF_PEDS0005) or whole genome sequencing (WGS; CLF_PEDS9001) on the primary kidney tumor tissues. In both patients, we confirmed the presence of sickle cell trait but also found tumor purity was <20%, which, like prior studies, prevented the identification of the fusion events (*Figure 1—figure supplement 1c*). This low tumor purity is attributable to the stromal desmoplasia seen in RMC (*Swartz et al., 2002*).

We then performed WES on the normal cell line (CLF_PEDS0005_N) or whole blood (CLF_PEDS9001) and compared it to the primary tumor cell lines (CLF_PEDS0005_T1 and CLF_PEDS9001_T) and metastatic cell lines (CLF_PEDS0005_T2A and CLF_PEDS0005_T2B). We found a low mutation frequency (1–3 mutations/mb; Materials and methods; *Figure 1a*) in the tumor cell lines similar to that of other pediatric cancers and cell lines such as MRT, ATRT and Ewing sarcoma (*Cibulskis et al., 2013*; *Johann et al., 2016*; *Wala et al., 2018*). We found that only the metastatic cell lines harbored mutations in *TP53* and *TPR* (Materials and methods; *Supplementary file 1*) (*Cibulskis et al., 2013*). Using copy number analysis, we confirmed the heterozygous loss of *SMARCB1*. In agreement with prior studies, we failed to find an identifiable mutation or deletion to account for the loss of the second *SMARCB1* allele with WES.

We used dual-color break apart FISH and found that a fusion event led to loss of the second *SMARCB1* allele as seen in prior studies (Materials and methods; *Supplementary file 2*) (*Calderaro et al., 2016*; *Carlo et al., 2017*). We then performed WGS to assess for structural variations that would not be captured by WES to elucidate the breakpoint of the rearrangement in *SMARCB1* (*Figure 1b*; Materials and methods). For CLF_PEDS0005_T1, we found a large deletion between *BCR* and *MYH9* which is predicted to lead to loss of one allele of *SMARCB1* (*Figure 1c*) along with a balanced translocation between intron 1 of *SMARCB1* to the intron region following the C-terminal end of *C1orf116*, yielding a non-functional allele (*Figure 1d* and *Supplementary file 3*). For CLF_PEDS9001_T1, we found a large deletion between *TTC28* and *VPREB* that would lead to loss of one allele of *SMARCB1* (*Figure 1e*) along with a balanced translocation that leads to fusion of intron 10 of *PLEKHA5* to intron 6 of *SMARCB1* (*Figure 1f* and *Supplementary file 4*). Both translocations involved inactivation of the C-terminal end of SMARCB1 (*Figure 1—figure supplement 1d*). We confirmed these findings by Sanger sequencing of the breakpoint, by qRT-PCR and by immunoblotting to demonstrate loss of SMARCB1 expression (*Supplementary file 5*, *Figure 1—figure supplement 1e–f*; Materials and methods). We then assessed previously identified breakpoint SMARCB1 sequences with the breakpoints identified in this study and failed to find alignments, fragile sites or other repetitive DNA elements that were shared amongst these sequences (*Figure 1—figure supplement 1g*). Taken together, we have developed in vitro cell line models from two patients with RMC which faithfully recapitulate known genomics of this disease.

## Patient-derived models of RMC are similar to SMARCB1 deficient cancers

We performed RNA-sequencing and transcriptomic profiling to compare the RMC models to other renal tumors or tumors that harbor loss of *SMARCB1*. Specifically, we compared the Therapeutically

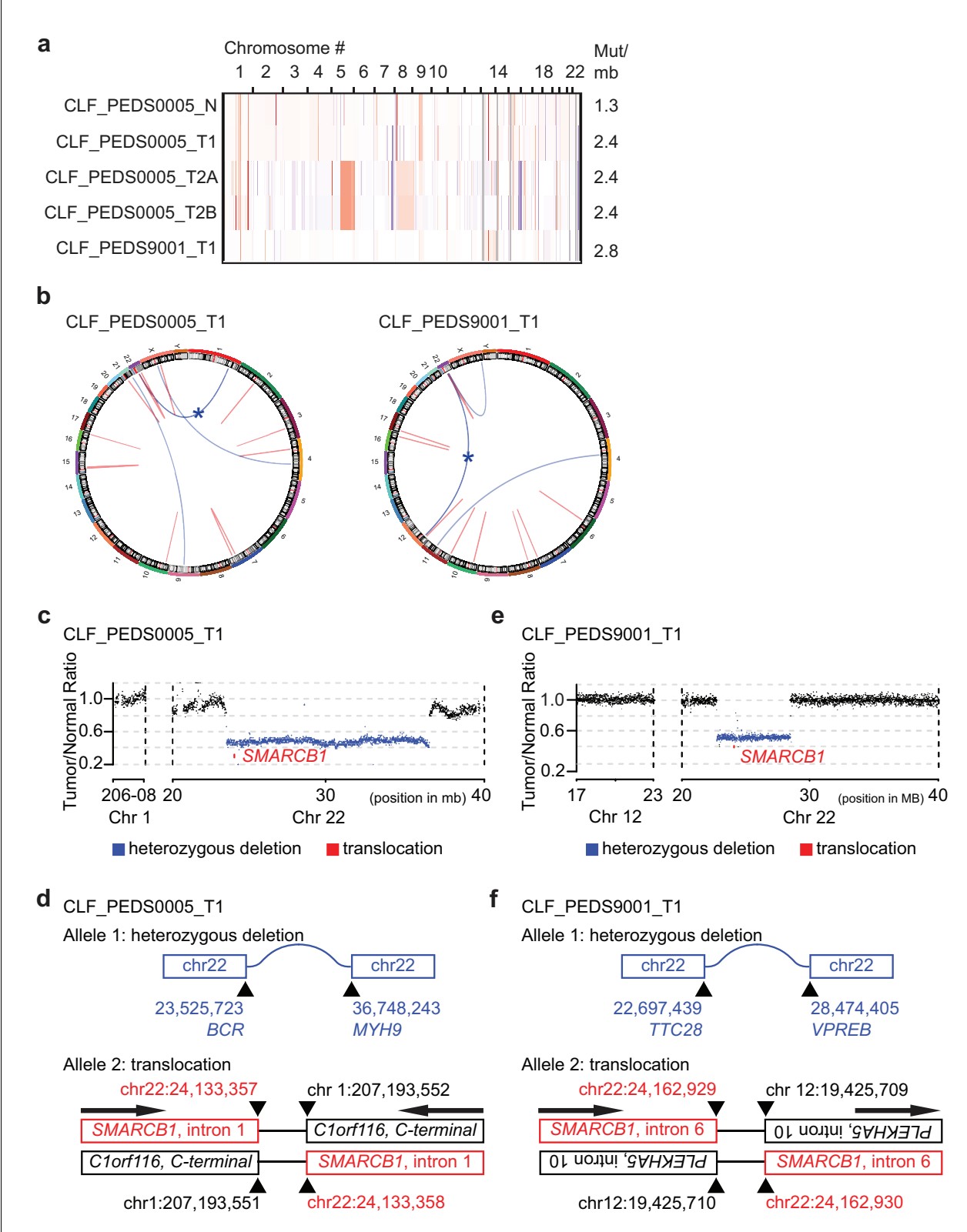

**Figure 1.** Patient-derived models of RMC have quiet genomes and are driven by intronic translocations. (a) Copy number analysis of RMC models from WES identifies heterozygous loss of alleles particularly at 22q where *SMARCB1* resides or low-level gains in primary and metastatic tumors. Red indicates degree of genomic amplification while blue indicates degree of genomic loss. Rates of mutations per megabase are consistent with patients with RMC or other pediatric cancers such as rhabdoid tumor. (b) Circos plots from WGS to represent structural alterations seen in RMC cell lines. Red

*Figure 1 continued on next page*

*Figure 1 continued*

indicates a deletion. All deletions are located in the introns. Blue arcs indicate fusions identified with SvABA v0.2.1 (*Wala et al., 2018*). Blue star indicates a *SMARCB1* rearrangement. (c) Read counts from WGS identify single copy deletion of *SMARCB1* in CLF_PEDS0005_T1 indicated by a Tumor/Normal ratio of approximately 0.5 where *SMARCB1* is located. (d) Second allele of *SMARCB1* is lost by a balanced translocation occurring in intron 1 of *SMARCB1* and fuses to the C-terminal end of C1orf116 in chromosome one in CLF_PEDS0005_T1. (e) Read counts from WGS identify single copy deletion of *SMARCB1* in CLF_PEDS9001_T1. (f) Second allele of *SMARCB1* is lost by a balanced translocation occurring in intron 6 of *SMARCB1* and fuses to the anti-sense intron 10 of PLEKHA5 in chromosome 12 in CLF_PEDS9001_T1.

DOI: https://doi.org/10.7554/eLife.44161.003

The following figure supplement is available for figure 1:

**Figure supplement 1.** Patient-derived RMC models are reflective of known biology and are faithful models of the primary tumors.
DOI: https://doi.org/10.7554/eLife.44161.004

Applicable Research to Generate Effective Treatments (TARGET) RNA-sequencing data from pediatric renal tumors (e.g. Wilms Tumor, Clear Cell Sarcoma of the Kidney, and Malignant Rhabdoid Tumor) or normal kidney tissues with the RMC models using t-distributed stochastic neighbor embedding (tSNE) (Materials and methods). The normal cell line, CLF_PEDS0005_N, clustered with TARGET normal kidney tissues and RMC cell lines from both patients clustered with the TARGET Rhabdoid Tumor samples (*Figure 2a*). These observations showed that these RMC cell lines share expression patterns with patients with MRTs.

To determine if the RMC cell lines clustered separately among other SMARCB1-deficient cancers, we performed gene expression analysis of our RMC models and compared them to publicly available datasets of MRT cell lines, patients with RMC, MRT or ATRT or synovial sarcoma (a cancer driven by the fusion oncoprotein SSX-S18 which displaces SMARCB1; Materials and methods) (*Barretina et al., 2012*; *Calderaro et al., 2016*; *Han et al., 2016*; *Johann et al., 2016*; *Richer et al., 2017*). Using tSNE, we found that the RMC cell lines closely mapped to a French cohort of RMC, MRT and ATRT patients (*Figure 2b*). These observations demonstrated that RMC cell lines and SMARCB1 deficient patients express similar gene expression programs.

We then assessed the consequences of re-expressing SMARCB1. Specifically, we generated doxycycline-inducible open reading frame (ORF) vectors harboring *SMARCB1* and stably infected our RMC models, G401 (MRT cell line) and HA1E (SMARCB1 wild-type immortalized epithelial kidney cell line) (*Hahn et al., 1999*). We confirmed that the addition of doxycycline used in our studies did not affect the proliferation of the parental cell lines (Materials and methods and *Figure 2—figure supplement 1a–b*).

We used these inducible cell lines to assess the biochemical stability of the SWI/SNF complex by using 10–30% glycerol gradient sedimentation followed by SDS-PAGE (*Figure 2c–d* and *Figure 2—figure supplement 1c*). In HA1E SMARCB1 wild-type cells, the SWI/SNF complex members SMARCB1, ARID1A and SMARCA4 were robustly expressed at higher molecular weights (e.g. fractions 13–16). In G401 SMARCB1 deficient cells, the SWI/SNF complex is smaller and seen at lower molecular weights (e.g. fractions 10–14). Furthermore, expression of SWI/SNF complex members was modestly decreased in G401, consistent with our prior studies (*Nakayama et al., 2017*; *Wang et al., 2017*). In our RMC cell lines, we found that the majority of ARID1A and SMARCA4 was observed in fractions 11–13 similar to what we found in the MRT cell line G401. In addition, we found increased expression and a shift of ARID1A and SMARCA4 to larger fractions 13–15 upon re-expression of SMARCB1 in the RMC lines similar to what we observed in the SMARCB1 wild-type HA1E cell line. We concluded that the composition of the SWI/SNF complex is similar between RMC and other SMARCB1 deficient cancers.

We then used the inducible cell lines to measure the consequence of SMARCB1 re-expression on the viability of the cells. We also generated cell lines with inducible expression of a LacZ control to compare with re-expression of SMARCB1. In the HA1E SMARCB1 wild type cells, we found no significant difference in viability between induction of SMARCB1 versus induction of LacZ using direct counting of viable cells (*Figure 2e*). In contrast, induction of SMARCB1 in the MRT G401 SMARCB1 deficient cells decreased the number of viable cells by 41%. Similar to G401, we found that re-expression of SMARCB1 in each of the RMC models led to significant decreases in cell viability (37–62%) (*Figure 2e*; *Figure 2—source data 1*). These observations suggest loss of SMARCB1 is required for the proliferation and viability of RMC cells.

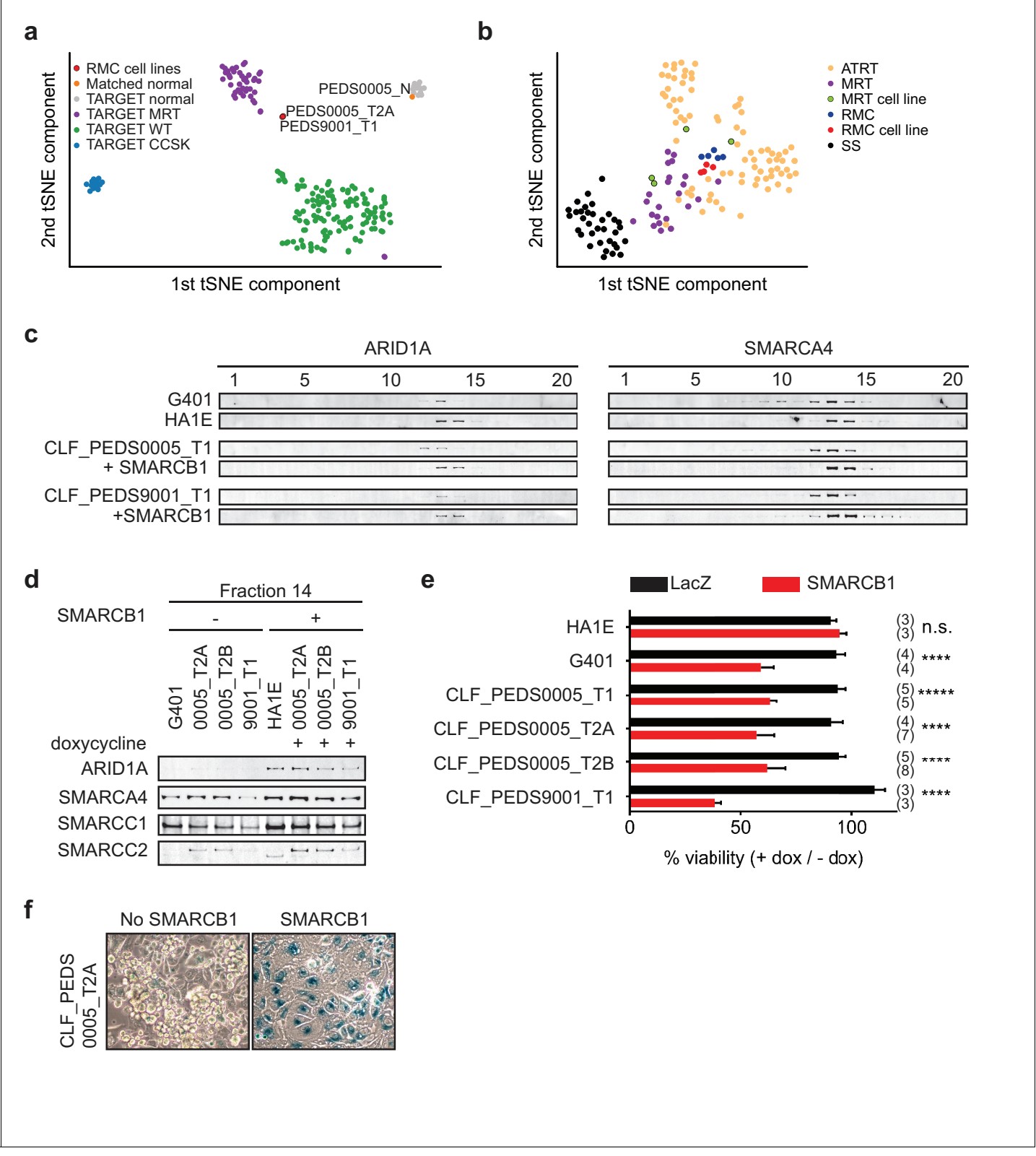

**Figure 2.** Patient-derived models of RMC are functionally similar to SMARCB1 deficient cancers. (a) Using tSNE, gene expression by RNA-sequencing from patients with pediatric renal tumors profiled in TARGET such as clear cell sarcoma of the kidney (CCSK), malignant rhabdoid tumor (MRT) or Wilms tumor (WT) was compared with patient-derived models of RMC. RMC cell lines (red) cluster with rhabdoid tumor samples (purple). The normal cell line (orange) (CLF_PEDS0005_N) clusters with other normal kidneys profiled in TARGET. (b) tSNE analysis of gene-expression array data shows RMC

*Figure 2 continued on next page*

*Figure 2 continued*

cell lines (red) clustering with RMC patients (blue) and these cluster with other MRT (purple) or ATRT (yellow) samples. However, these do not cluster as closely with synovial sarcomas (black). (**c**) Glycerol gradients (10–30%) followed by SDS-PAGE analysis of rhabdoid tumor cell line G401, as compared to SMARCB1 wild-type cell line HA1E, show ARID1A is seen in higher fractions when SMARCB1 is expressed (left). Gradients were then performed on patient-derived models of RMC with doxycycline-inducible SMARCB1. A similar rightward shift of ARID1A occurs upon re-expression of SMARCB1. These same shifts occur with SMARCA4 (right). These experiments are representative of at least two biological replicates. (**d**) Fraction 14 of the glycerol gradients shows a modest increase in SWI/SNF complex members, SMARCC1, SMARCC2, SMARCA4 and ARID1A in HA1E, a SMARCB1 wild type cell line. When SMARCB1 is re-expressed in G401 and RMC lines, a similar pattern is seen. Images are representative of 2 biological replicates. (**e**) Using cell lines with stably transfected and inducible SMARCB1, cell viability was assessed with or without expression of SMARCB1 over 8 days. There is no significant difference in SMARCB1 wild type cell line, HA1E. Re-expression of SMARCB1 leads to significant decreases in cell viability as compared to LacZ control in SMARCB1 deficient cancer cell lines G401, CLF_PEDS9001_T, and CLF_PEDS0005. Error bars are standard deviations based on number of samples in parentheses. (**f**) CLF_PEDS0005_T2A cell line shows signs of senescence following re-expression of SMARCB1. Images representative of 3 biological replicates.

DOI: https://doi.org/10.7554/eLife.44161.005

The following source data and figure supplement are available for figure 2:

**Source data 1.** Source data for *Figure 2e*.

DOI: https://doi.org/10.7554/eLife.44161.006

**Figure supplement 1.** Kinetics of re-expression of SMARCB1 in patient-derived RMC models.

DOI: https://doi.org/10.7554/eLife.44161.007

Since MRT cell lines arrest and senesce when SMARCB1 is re-expressed (*Betz et al., 2002*), we looked for evidence of senescence by staining for senescence-associated acidic β-galactosidase in the RMC cells when SMARCB1 was re-expressed. Following 7 days of SMARCB1 or LacZ re-expression, we stained the cells for β-galactosidase (Materials and methods). We failed to observe cells expressing β-galactosidase upon expression of LacZ in the RMC cells, but when SMARCB1 was expressed, we found 44.6% (±17%) of the RMC cells stained for β-galactosidase (*Figure 2f* and *Figure 2—figure supplement 1d*). These studies showed that re-expression of SMARCB1 in RMC cells may also lead to senescence.

We then assessed what genes were differentially expressed upon SMARCB1 re-expression in the RMC and MRT cell lines as another way to assess the similarity between these two cancers. We performed RNA-sequencing on the doxycycline-induced SMARCB1 RMC cell lines and compared them to the uninduced cell lines or doxycycline-induced LacZ cell lines. We then re-analyzed our previously published studies of MRT cells with SMARCB1 re-expression and compared them to our RMC cells (*Wang et al., 2017*). We found 1719 genes to be significantly different (false discovery rate of <0.25) in the RMC cells and 2735 genes in MRT cells. We identified 527 genes that significantly overlapped between the RMC and MRT cell lines (hypergeometric p-value less than 4.035e-63; *Supplementary file 6*). We compared this group of genes to the genes differentially expressed between MRT tumor and normal tissues from TARGET (n = 6,311). We identified 257 genes that overlapped with the 527 significantly differentially expressed genes induced by re-expression of SMARCB1 (*Supplementary file 6*).

Using this list of 257 genes, we performed gene ontology (GO)-based Gene Set Enrichment Analysis (GSEA) (*Subramanian et al., 2005*) and identified significantly enriched genes sets (q-value <0.1), including those related to the cell cycle and the ubiquitin-proteasome system (UPS). We then analyzed the kinetics by which these gene expression changes occur after SMARCB1 was re-expressed. Specifically, we analyzed 5 genes of these 257 genes that are implicated in regulation of the G1/S (*RRM2*, *TOP2A*) or G2/M (*PLK1*, *CCNB1*, *UBE2C*) phases of the cell cycle. For *PLK1* and *CCNB1*, we observed a gradual decrease in expression over the course of 120 hr whereas *RRM2*, *UBE2C* and *TOP2A* exhibited a more profound decrease in expression after the first 24 hr and then a modest decrease over the following 96 hr (*Figure 2—figure supplement 1e–f*).

These findings confirm that changes in the transcriptome following SMARCB1 re-expression in RMC cell lines are similar to other SMARCB1 deficient cancer cell lines. In sum, these observations indicate that the RMC cell lines are functionally similar to those derived from other SMARCB1 deficient cancers.

## RNAi and CRISPR-Cas9 loss-of-function screens and small-molecule screens in RMC models identify proteasome inhibition as a vulnerability

MRT, ATRT and RMC are aggressive and incurable cancers. We performed genetic (RNAi and CRISPR-Cas9) and pharmacologic screens to identify druggable targets that would decrease proliferation or survival for these cancers. Specifically, we used the Druggable Cancer Targets (DCT v1.0) libraries and focused on targets that were identified by suppression with RNAi, gene deletion with CRISPR-Cas9-based genome editing, and perturbation by small molecules (*Hong et al., 2016*; *Seashore-Ludlow et al., 2015*). We accounted for off-target effects in the RNAi screens by using seed controls for each shRNA.

We performed these three orthogonal screens on both metastatic models of RMC, CLF_PEDS0005_T2A and CLF_PEDS0005_T2B (*Figure 3a*). We introduced the shRNA DCT v1.0 lentiviral library into these two cell lines and evaluated the abundance of the shRNAs after 26 days using massively parallel sequencing (Materials and methods). We confirmed depletion of known common essential genes such as *RPS6* (*Figure 3—figure supplement 1a*). We then analyzed the differential abundance between the experimental and seed control shRNAs to collapse individual shRNAs to consensus gene dependencies with RNAi Gene Enrichment Ranking (RIGER) (*Luo et al., 2008*). Of 444 evaluable genes, 72 genes scored with a RIGER p-value<0.05 in CLF_PEDS0005_T2A and 74 genes scored in CLF_PEDS0005_T2B.

In parallel, we introduced the CRISPR-Cas9 DCT v1.0 lentiviral library to determine the differential representation of the CRISPR-Cas9 sgRNAs between 6 and 23 days to identify genes depleted or enriched in this screen by massively parallel sequencing (Materials and methods). We confirmed that the distribution of sgRNAs among biological replicates was highly correlated (*Figure 3—figure supplement 1b–e*). When compared to the controls, there was significant depletion of essential genes such as *RPS6* (*Figure 3—figure supplement 1f–g*). We used RIGER to collapse the individual sgRNAs to consensus gene dependencies and found 124 genes (of a total of 445 evaluable genes) and 136 genes (of a total of 445 evaluable genes) with a RIGER p-value<0.05 in CLF_PEDS0005_T2A and CLF_PEDS0005_T2B cell lines, respectively.

We performed a small-molecule screen using a library of 440 compounds that have known targets in the RMC cells (CLF_PEDS0005_T2A and CLF_PEDS0005_T2B) (*Hong et al., 2016*). This library includes 72 FDA approved compounds, 100 compounds in clinical trials and 268 probes based on our prior studies. We calculated an area under the curve (AUC) based on an 8-point concentration range and considered AUCs < 0.5 as significant. Of the evaluable compounds, 75 (18%) compounds significantly decreased cell viability in CLF_PEDS0005_T2A and 82 (20%) compounds significantly decreased cell viability in CLF_PEDS0005_T2B.

We then looked for genes or targets of the small molecules that scored in all three of the RNAi, CRISPR-Cas9 and small-molecule screens. We identified 21 genes in CLF_PEDS0005_T2A and 27 genes in CLF_PEDS0005_T2B (*Supplementary file 7*) of which 19 genes scored in both screens (*Figure 3a*). Among the 19 genes were components of the ubiquitin-proteasome system (e.g. *PSMB1*, *PSMB2*, *PSMB5*, *PSMD1*, *PSMD2*, and *CUL1*), regulators of the cell cycle (*CDK1*, *CDK6*, *KIF11* and *PLK1*) and genes involved in nuclear export (*KPNB1* and *XPO1*).

To eliminate small molecules and targets that affect normal renal tissue, we screened the normal cell line (CLF_PEDS0005_N) with the small-molecule library. We calculated the robust Z-scores for these screens in relationship to the Cancer Cell Line Encyclopedia (CCLE) to normalize the responses to various compounds (*Barretina et al., 2012*; *Rees et al., 2016*; *Seashore-Ludlow et al., 2015*). We then compared the results of this small-molecule screen with the RMC cancer cell lines (CLF_PEDS0005_T2A and CLF_PEDS0005_T2B). We found that the tumor cells were differentially sensitive (up to two standard deviations) upon treatment with proteasome inhibitors, bortezomib and MLN2238, when compared to the normal cell line (*Figure 3b*; *Figure 3—source data 1*). These findings suggest that the vulnerability to proteasome inhibition may be dependent on loss of SMARCB1.

## Validation of proteasome inhibition as a specific therapy in SMARCB1 deficient cancers

To validate the dependency of SMARCB1 deficient tumors to the ubiquitin-proteasome system, we assessed the consequences of inhibiting proteasome function on survival of the primary tumor cell

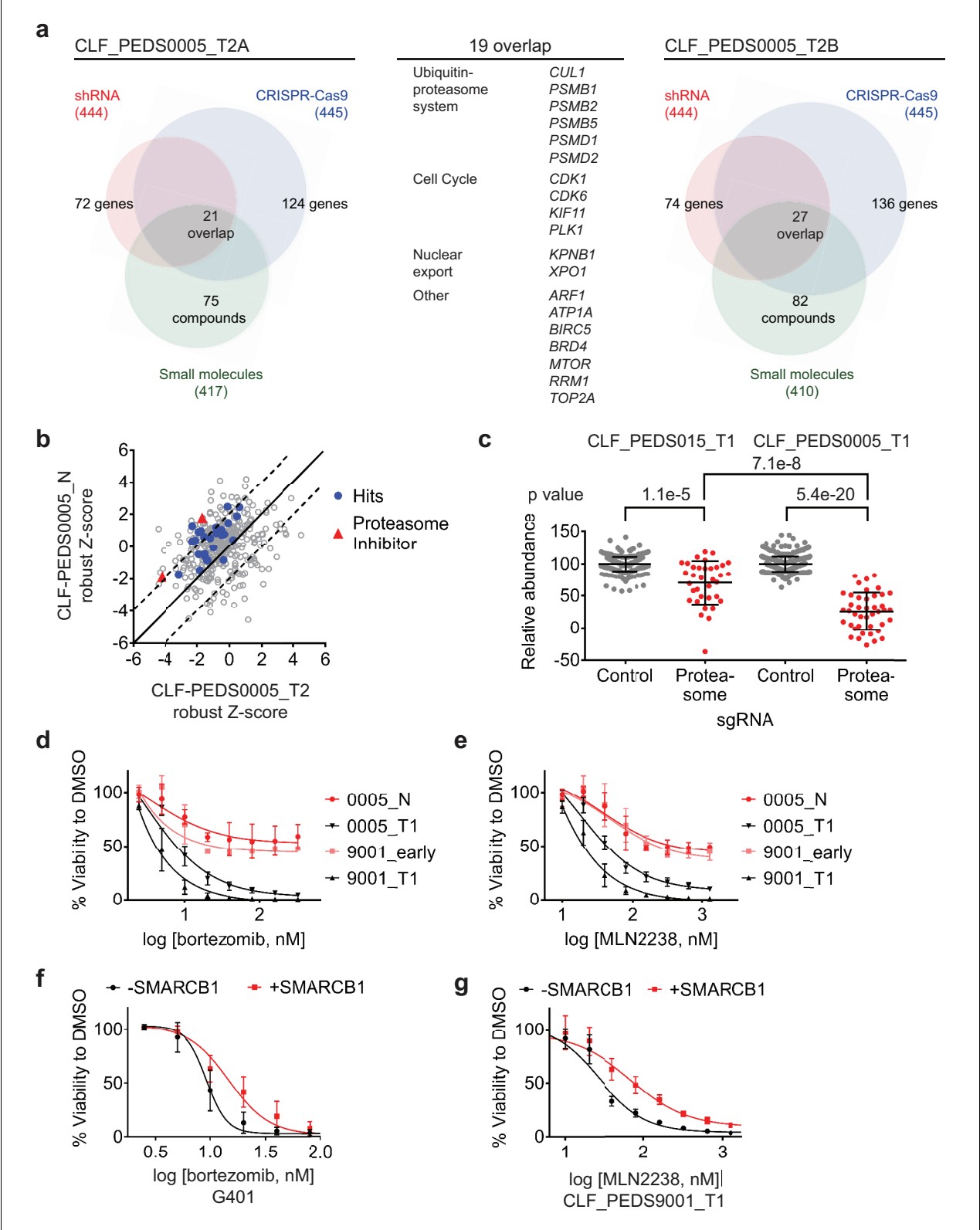

**Figure 3.** Functional genomic screens identify inhibition of the proteasome as a vulnerability in RMC. (a) Left: RNAi suppression of 444 evaluable genes (red) identifies 72 genes that when suppressed caused a significant viability loss in CLF_PEDS0005_T2A. Genomic indels created by CRISPR-Cas9 in 445 evaluable genes (blue) identify 124 genes that cause a viability loss. RNAi and CRISPR-Cas9 screens were performed in biological replicates. Small-molecule screen (performed in technical replicates) with 417 evaluable compounds (green) identifies 75 compounds that lead to significant viability loss. *Figure 3 continued on next page*

*Figure 3 continued*

21 genes overlap across these three screens. Right: The same screens were performed with CLF_PEDS0005_T2B and 27 genes were found to be significantly depleted when suppressed by RNAi, genomically deleted by CRISPR-Cas9 or inhibited when treated with a small molecule. Center: List of 19 genes that overlapped between functional screens from CLF_PEDS0005_T2A and CLF_PEDS0005_T2B can be categorized into genes involving the ubiquitin-proteasome system, cell cycle and nuclear export (***Supplementary file 7***). (b) Comparison of Z-score normalized small-molecule screens between CLF_PEDS0005_T2 and CLF_PEDS0005_N (normal isogenic cell line). Small molecules targeting the genes identified in ***Figure 3a*** are either in red (proteasome inhibitors) or blue (other hits). Each dot is representative of the average of two technical replicates. (c) Relative log2 fold change in abundance from CRISPR-Cas9 screens between sgRNA controls (grey) and genes in the DCT v1.0 screen involving the proteasome (red). Data is taken at 23 days following selection and compared to an early time point. As compared to the undifferentiated sarcoma cell line CLF_PEDS015_T1, inhibition of the proteasome subunits leads to a more profound viability loss as compared with controls. Each dot is representative of a minimum of 2 biological replicates. (d) Short term cultures of the normal cell line (CLF_PEDS0005_N) or early passage of the heterogenous cell line (CLF_PEDS9001_early) were compared for assessment of viability to the primary tumor cell lines following treatment with bortezomib. Two-tailed t-test p-value=0.008 for PEDS0005_T1 and two-tailed t-test p-value=4.76e-5 for PEDS9001_T1. Error bars represent standard deviations from two biological replicates. (e) Short term cultures of the normal cell line (CLF_PEDS0005_N) or early passage of the heterogenous cell line (CLF_PEDS9001_early) were compared for assessment of viability to the primary tumor cell lines following treatment with MLN2238. Error bars represent standard deviations from two biological replicates. (f) Re-expression of SMARCB1 in G401 leads to a rightward shift in the dose-response curve with bortezomib compared with uninduced cells. Error bars represent standard deviations from two biological replicates. (g) Re-expression of SMARCB1 in CLF_PEDS9001_T leads to a rightward shift in the dose-response curve with MLN2238 compared with uninduced cells. Error bars represent standard deviations from three biological replicates.

DOI: https://doi.org/10.7554/eLife.44161.008

The following source data and figure supplement are available for figure 3:

**Source data 1.** Source data for ***Figure 3b***.
DOI: https://doi.org/10.7554/eLife.44161.010

**Figure supplement 1.** Functional genomic screens identify proteasome inhibition as a vulnerability in RMC and other SMARCB1 deficient cancers.
DOI: https://doi.org/10.7554/eLife.44161.009

line, CLF_PEDS0005_T1, by deleting components of the proteasome with CRISPR-Cas9. We compared these findings with a model of undifferentiated sarcoma, CLF_PEDS015T, that does not harbor mutations in *SMARCB1* (***Hong et al., 2016***). We scaled the results based on the non-targeting sgRNA negative controls and positive controls targeting *RPS6*, a common essential gene (***Hart et al., 2015***). Compared to the control sgRNAs, there was an average decrease of 29% in viability in CLF_PEDS015T while there was an average decrease of 74% in CLF_PEDS0005_T1 (***Figure 3c***; Materials and methods). Although deletion of the proteasome members affected proliferation in all of the models (two tailed t-test p=1.1e-5 for CLF_PEDS015T and p=5.4e-20 for CLF_PEDS0005_T1), we found that suppression of proteasome components affected the RMC model CLF_PEDS0005_T1 to a statistically greater degree (two tailed t-test p=7.1e-8). We subsequently validated that gene deletion by CRISPR-Cas9 of *PSMB5*, one of the primary targets of proteasome inhibitors, in the CLF_PEDS0005_T2A and CLF_PEDS9001_T1 cell lines led to decreased viability (***Figure 3—figure supplement 1h–i***).

We then determined whether this vulnerability to proteasome inhibition was specific to the loss of SMARCB1. We treated the normal cell line, CLF_PEDS0005_N, and an early passage of CLF_PEDS9001_T1 while it was a heterogenous population and retained *SMARCB1* (***Figure 3—figure supplement 1j***) with bortezomib or the second-generation proteasome inhibitor, MLN2238. We observed significantly decreased sensitivity to the proteasome inhibitors in the *SMARCB1* retained isogenic cell lines as compared to the *SMARCB1* deficient cell lines (***Figure 3d–e***). We then treated our SMARCB1-inducible RMC and MRT cell lines with DMSO, bortezomib or MLN2238. Re-expression of SMARCB1 led to a decrease in sensitivity to bortezomib or MLN2238 as compared to the isogenic SMARCB1 deficient lines (***Figure 3f–g***). The observed differential resistance to SMARCB1 re-expression was between 2–3-fold with either bortezomib or MLN2238 (***Figure 4—figure supplement 1a–b***). We concluded that re-expression of SMARCB1 partially rescued the sensitivity of MRT or RMC cell lines to proteasome inhibition.

We then compared the results of small-molecule screens performed in SMARCB1-deficient cancer cell lines in CCLE to the rest of the CCLE cell lines (n = 835). We found that SMARCB1-deficient cell lines were significantly more sensitive (two-tailed t-test p-value=0.011) to treatment with MLN2238 than non-multiple myeloma CCLE cell lines (***Figure 4a***; ***Figure 4—source data 1***) (***Rees et al., 2016***; ***Seashore-Ludlow et al., 2015***). The degree of sensitivity was similar to that of multiple myeloma cell

lines which are known to be sensitive to proteasome inhibition (*Dimopoulos et al., 2016*). These findings confirm that SMARCB1 deficient cell lines are selectively vulnerable to proteasome inhibition.

We then performed in vitro studies to confirm the findings from these high throughput small molecule screens. We treated an additional 6 SMARCB1 deficient cell lines (4 MRT and 2 ATRT) with bortezomib. We compared these results to H2172, a lung cancer cell line that was not sensitive to proteasome inhibition in CCLE small-molecule screens, and RPMI8226, an established multiple myeloma cell line that is responsive to proteasome inhibition (*Hideshima et al., 2001*). We found that our SMARCB1 deficient cell lines exhibited single digit nanomolar sensitivity to proteasome inhibition similar to that observed in the multiple myeloma cell line RPMI8226 (*Figure 4—figure supplement 1c*). In contrast, we found that the IC50 in H2172 was at least 3-fold higher. Since there are no SMARCB1 wildtype pediatric kidney cancer cell lines in CCLE, we compared the sensitivity to bortezomib or MLN2238 in our RMC models with wildtype SMARCB1 patient-derived Wilms tumor cell line, CLF_PEDS1012_T (*Figure 4b–c* and *Figure 3—figure supplement 1j*). We found that CLF_PEDS1012_T was more resistant to proteasome inhibition as compared to our RMC models and MRT cell line, G401. These findings suggest that SMARCB1-deficient cells are more sensitive to proteasome inhibition.

## Proteasome inhibition leads to cell cycle arrest in G2/M and subsequent cell death

We then studied how SMARCB1 loss leads to a dependency on the ubiquitin-proteasome system. Since activation of c-MYC has been observed in SMARCB1-deficient cancers (*Cheng et al., 1999*; *Genovese et al., 2017*), we assessed how c-MYC levels are altered upon proteasome inhibition. Compared to RPMI8226, a multiple myeloma cell line that relies on c-MYC for survival (*Tagde et al., 2016*), we failed to observe suppression of c-MYC protein levels following bortezomib or MLN2238 treatment (*Figure 4—figure supplement 1d*). We then assessed c-MYC expression levels in the G401, CLF_PEDS9001T, and CLF_PEDS00005_T1 cell lines following treatment with MLN2238 and found that c-MYC levels were increased after MLN2238 treatment (*Figure 4—figure supplement 1e*). In our models of RMC and MRT, these findings suggest proteasome inhibition does not lead to suppression of c-MYC.

We then assessed if the SWI/SNF complex is altered upon treatment with a proteasome inhibitor. We treated uninduced and induced SMARCB1 cells with DMSO or a proteasome inhibitor. We failed to see a consistent significant change in total or nuclear protein when immunoblotting for SWI/SNF complex members (SMARCE1, SMARCD1, SMARCD1, SMARCC1, SMARCC2, SMARCA4 and ARID1A) other than increases in SMARCB1 levels upon doxycycline treatment (*Figure 4—figure supplement 2a–b*). These findings suggest that inhibition of the proteasome in SMARCB1 deficient cancers and its subsequent resistance upon SMARCB1 expression does not alter the total or nuclear levels of SWI/SNF complex members.

ER stress has been implicated as a mechanism by which proteasome inhibitors act on multiple myeloma cells (*Obeng et al., 2006*). We saw an increase in protein expression of markers of ER stress, GRP78 and IRE1$\alpha$, following treatment with MLN2238 (*Figure 4—figure supplement 2c*). Upon re-expression of SMARCB1 and subsequent treatment with a proteasome inhibitor, we did not see changes in either GRP78 or IRE1$\alpha$ protein levels (*Figure 4—figure supplement 2c*). These observations suggest that although ER stress markers are elevated upon proteasome inhibition in SMARCB1-deficient cell lines, they are not rescued by SMARCB1 re-expression.

We then performed GO-based GSEA (*Subramanian et al., 2005*) on the 527 significantly altered genes upon re-expression of SMARCB1 (*Supplementary file 6*) to identify classes of gene function enriched in this group of genes. We identified numerous gene sets that involved the ubiquitin-proteasome system (*Supplementary file 8*). We then performed RNA-sequencing on G401 and the RMC cell lines treated with DMSO or MLN2238. We identified 1758 genes which were significantly (FDR < 0.1) up- or down-regulated upon treatment with MLN2238 (*Supplementary file 9*). We compared the 527 differentially expressed genes identified upon re-expression of SMARCB1 with the 1758 differentially expressed genes identified upon treatment with MLN2238 and identified 92 genes which overlapped. Of these genes, we identified 63 genes which were differentially expressed with re-expression of SMARCB1 and were inversely differentially expressed with treatment with MLN2238 (*Figure 4d*; *Figure 4—source data 2*). From this refined gene set, we performed GO-

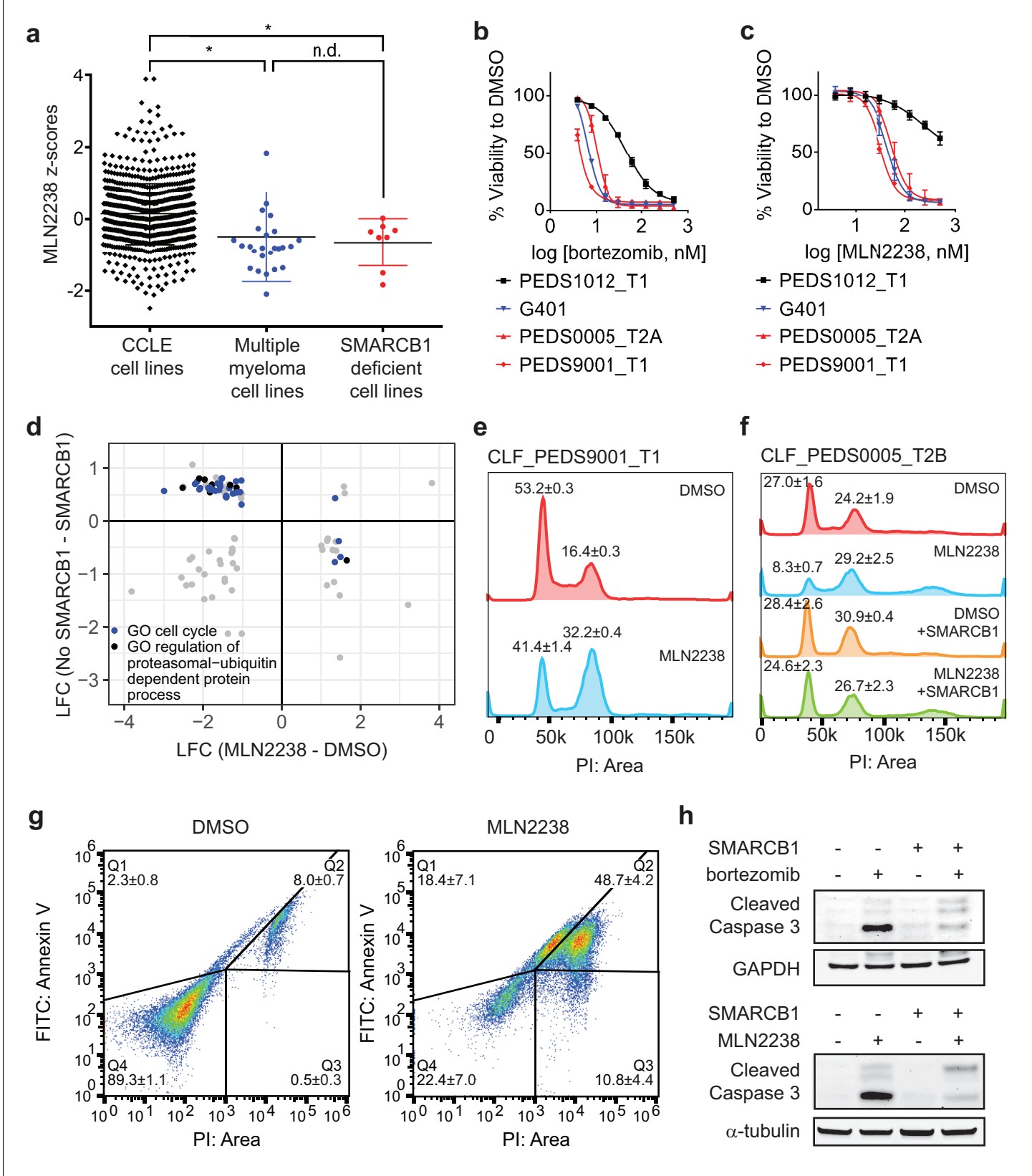

**Figure 4.** Proteasome inhibitors such as MLN2238 are specific to SMARCB1 deficient cancers and lead to cell cycle arrest and programmed cell death. (a) Multiple myeloma cell lines and SMARCB1-deficient lines are similarly sensitive to proteasome inhibitor, MLN2238. Both are significantly different

*Figure 4 continued on next page*

 Research article

Cancer Biology | Human Biology and Medicine

Figure 4 continued

from other CCLE cell lines based on Z-score normalized sensitivity. * two-tailed t-test p-value<0.05. n.d. no difference. (**b**) Early passage Wilms tumor (SMARCB1 wild-type) cell line CLF_PEDS1012_T1 is not as sensitive to treatment with bortezomib compared with RMC and MRT cell lines. Error bars represent standard deviations following at least two biological replicates. (**c**) Early passage Wilms tumor (SMARCB1 wild-type) cell line CLF_PEDS1012_T1 is not as sensitive to treatment with MLN2238 compared with RMC and MRT cell lines. Error bars represent standard deviations following at least two biological replicates. (**d**) Analysis of differentially expressed genes when SMARCB1 was re-expressed in SMARCB1 deficient cancers compared with differentially expressed genes when SMARCB1 deficient cancers were treated with 200 nM MLN2238. Gene sets enriched based on GO-based GSEA involved the cell cycle (blue) and regulation of the ubiquitin-proteasome system (black). (**e**) Treatment with proteasome inhibitor, MLN2238 at 200 nM for 24 hr leads to G2/M arrest in CLF_PEDS9001_T1. Values shown represent the percent of cells in G1 or G2/M. Error values shown are standard deviations from two biological replicates. (**f**) Treatment with MLN2238 at 200 nM for 24 hr leads to G2/M arrest in CLF_PEDS0005_T2B which can be prevented by re-expression of SMARCB1. Error values shown are standard deviations from two biological replicates. (**g**) Treatment of CLF_PEDS9001_T1 with MLN2238 at 200 nM for 48 hr leads to increased frequency of cells with Annexin V/PI staining and PI only staining. Error values shown are standard deviations from two biological replicates. (**h**) G401 cells stably infected with inducible SMARCB1 treated with either bortezomib at 15 nM or MLN2238 at 200 nM induce cleaved caspase-3 compared with DMSO controls after 24 hr. When SMARCB1 is re-expressed, cleaved caspase-3 levels are decreased compared to uninduced cell lines. Blots are representative of a minimum of 2 biological replicates.

DOI: https://doi.org/10.7554/eLife.44161.011

The following source data and figure supplements are available for figure 4:

**Source data 1.** Source data for *Figure 4a*.
DOI: https://doi.org/10.7554/eLife.44161.014
**Source data 2.** Source data for *Figure 4d*.
DOI: https://doi.org/10.7554/eLife.44161.015
**Figure supplement 1.** SMARCB1 deficient cell lines are sensitive to proteasome inhibition.
DOI: https://doi.org/10.7554/eLife.44161.012
**Figure supplement 2.** Treatment with proteasome inhibition does not lead to changes that can be rescued by SMARCB1 re-expression in the nuclear or cytoplasic levels of SWI/SNF complex members or ER stress proteins but is in part driven by cell cycle arrest in G2/M.
DOI: https://doi.org/10.7554/eLife.44161.013

based GSEA (*Subramanian et al., 2005*) and found significant enrichment (adjusted p-value ranging from 0 to 0.0061) in gene sets involving the cell cycle (*Supplementary file 10*).

We subsequently assessed the cell cycle by DNA content with cells treated with DMSO or MLN2238 for 24 hr as proteasome inhibitors have been found to cause a G2/M cell cycle arrest in lymphomas, colorectal carcinomas, hepatocellular carcinomas, and glioblastoma multiforme (*Augello et al., 2018*; *Bavi et al., 2011*; *Gu et al., 2017*; *Yin et al., 2005*). We observed a significant shift in cells to G2/M (two-tailed t-test p-value 0.0005; *Figure 4e*). Upon re-expression of SMARCB1, we saw that this phenotype was rescued (*Figure 4f*). By 48 hr, we saw a significant increase in markers of programed cell death such as Annexin V and PI positive cells (*Figure 4g*).

We found that treatment with a proteasome inhibitor led to increased cleaved caspase-3 levels in addition to changes to Annexin V and PI, suggesting that inhibition of the ubiquitin-proteasome system leads to programmed cell death (*Figure 4h*). We then asked whether restoration of SMARCB1 expression inhibited cleaved caspase-3 activation. We found that induction of cleaved caspase-3 was less pronounced when SMARCB1 was re-expressed (*Figure 4h*). These observations suggest that proteasome inhibitors initially lead to a SMARCB1-dependent G2/M cell cycle arrest and subsequent programmed cell death.

To assess whether RMC cells exhibit an increased proclivity to undergo cell death after cell cycle arrest, we asked whether treatment of SMARCB1-deficient cancers with cell cycle inhibitors led only to cell cycle arrest or arrest followed by cell death. Specifically, we used nocodazole, an anti-mitotic agent that disrupts microtubule assembly in prometaphase, and RO-3306, a CDK1 inhibitor which disrupts the CDK1-cyclin B1 interaction during metaphase (*Vassilev et al., 2006*; *Wolf et al., 2006*). We treated both G401 and CLF_PEDS9001_T with nocodazole or RO-3306 for 24 hr and observed accumulation of cells in G2/M as well as increased cyclin B1 and cleaved caspase-3 (*Figure 4—figure supplement 2d–e*) similar to what we observed after treatment with MLN2238. By 72 hr, we found that treatment with either nocodazole or RO-3306 induced cell death in the majority of cells (65–90%) (*Figure 4—figure supplement 2f*), similar to what we observed when we treated cells with MLN2238 (*Figure 4c*). These observations suggest that SMARCB1-deficient cell lines are susceptible to programmed cell death following treatment with a cell cycle inhibitor and that the cell cycle arrest observed after treatment with MLN2238 leads to programmed cell death.

## Proteasome inhibitor induced G2/M arrest is mediated in part by inappropriate cyclin B1 degradation driven by a dependency on UBE2C

We subsequently searched for genes related to the ubiquitin-proteasome system and SMARCB1 function. We defined a set of 204 genes that were upregulated when comparing the log2 fold change between SMARCB1 deficient cells and SMARCB1 re-expressed cells in RMC and MRT cell lines (*Supplementary file 6*). We then took this set of 204 genes and examined the Project Achilles (genome scale CRISPR-Cas9 loss of function screens) DepMap Public 18Q3 dataset (*Meyers et al., 2017*) to determine whether any SMARCB1 deficient cell lines required expression of these genes for survival. This dataset included loss of function screens from 485 cancer cell lines and included three ATRT SMARCB1-deficient cancer cell lines: COGAR359, CHLA06ATRT and CHLA266. We found that SMARCB1 deficient cancer cell lines required *UBE2C*, an ubiquitin-conjugating enzyme, for survival. We noted that these cell lines were in the top 5% of cell lines (n = 485) that required *UBE2C* for survival (empirical Bayes moderated t-test p-value=0.00016; *Figure 5a–b*; *Figure 5—source data 1*). These observations suggested that cancer cell lines that lack SMARCB1 were also dependent on UBE2C.

Since the 3 cell lines profiled were ATRTs, we validated that the RMC cell lines were also dependent on *UBE2C* for survival. We generated sgRNAs specific for *UBE2C* and assessed viability by cell counting following gene deletion. We saw a significant decrease in cell viability in SMARCB1 deficient cell lines as compared to urothelial carcinoma cell line, JMSU1 (SWI/SNF wild type), or non-small cell lung cancer cell line, A549 (*SMARCA4* mutant) (two-tailed t-test p-values 4.5e-5 and 4.6e-5; *Figure 5c* and *Figure 5—figure supplement 1a*).

UBE2C serves as the E2 enzyme which adds the first ubiquitin (Ub) to cyclin B1 for degradation (*Dimova et al., 2012*; *Grice et al., 2015*). Cyclin B1 degradation is required in G2/M at the end of metaphase to enter anaphase (*Chang et al., 2003*). Our integrated RNAi, CRISPR-Cas9 and small molecule screens identified that our RMC models required expression of *PLK1* and *CDK1*, genes involved in G2/M, for survival (*Figure 3a–b*), and prior studies have identified that inhibition of *PLK1* in ATRT or MRT cells leads to arrest in G2/M (*Alimova et al., 2017*; *Morozov et al., 2007*). Treatment of the RMC cell lines with MLN2238 led to accumulation of cyclin B1 as compared to cyclin D1 suggesting that MLN2238 inhibits degradation of cyclin B1 (*Figure 5d*). When we re-expressed SMARCB1, we found that cyclin B1 levels were unchanged upon MLN2238 treatment. Although APC/C serves as the E3 ligase for cyclin B1, genetic deletion of APC/C in Project Achilles showed that APC/C was an essential gene across all cancer cell lines. These findings suggest SMARCB1 deficient cancer cells require UBE2C expression for survival, in part by regulating cyclin B1 stability.

## Effects of proteasome inhibition in vivo

These studies identify a lethal interaction between suppressing the UPS and SMARCB1-deficient cancers in vitro. The doses used in this study were based on in vitro studies of multiple myeloma or lymphoma cell lines (*Chauhan et al., 2011*; *Garcia et al., 2016*; *Hideshima et al., 2003*; *Hideshima et al., 2001*). For patients with primary or refractory multiple myeloma, use of proteasome inhibitors has led to significant clinical responses (*Jagannath et al., 2004*; *Moreau et al., 2016*; *Richardson et al., 2005*; *Richardson et al., 2003*). We reasoned that if our SMARCB1 deficient cancers were susceptible to proteasome inhibitors at similar in vitro dosing, we would also see similar in vivo responses. We first determined whether these doses led to proteasome inhibition by assessing the ability of these cell lines to cleave Suc-LLVY-aminoluciferin. We found that treatment of the SMARCB1 deficient cell lines with either bortezomib or MLN2238 led to inhibition of the proteasome to a similar extent observed when the multiple myeloma cell line RPMI8226 was treated (*Figure 5—figure supplement 1b*; Materials and methods). We also simulated the pharmacodynamics of proteasome inhibitors in vivo by treating cells in vitro with a pulse dose of proteasome inhibitors as has been performed in multiple myeloma and chronic myeloid leukemia cell lines (*Crawford et al., 2014*; *Kuhn et al., 2007*; *Shabaneh et al., 2013*). We found that upon treatment with MLN2238, SMARCB1 deficient cells arrested in G2/M, which led to cell death as measured by Annexin V/PI staining and led to accumulation of cyclin B1 similar to treatment with a continuous dose of MLN2238 (Materials and methods; *Figure 5—figure supplement 1c–f* and *Figure 5—figure supplement 2a–c*).

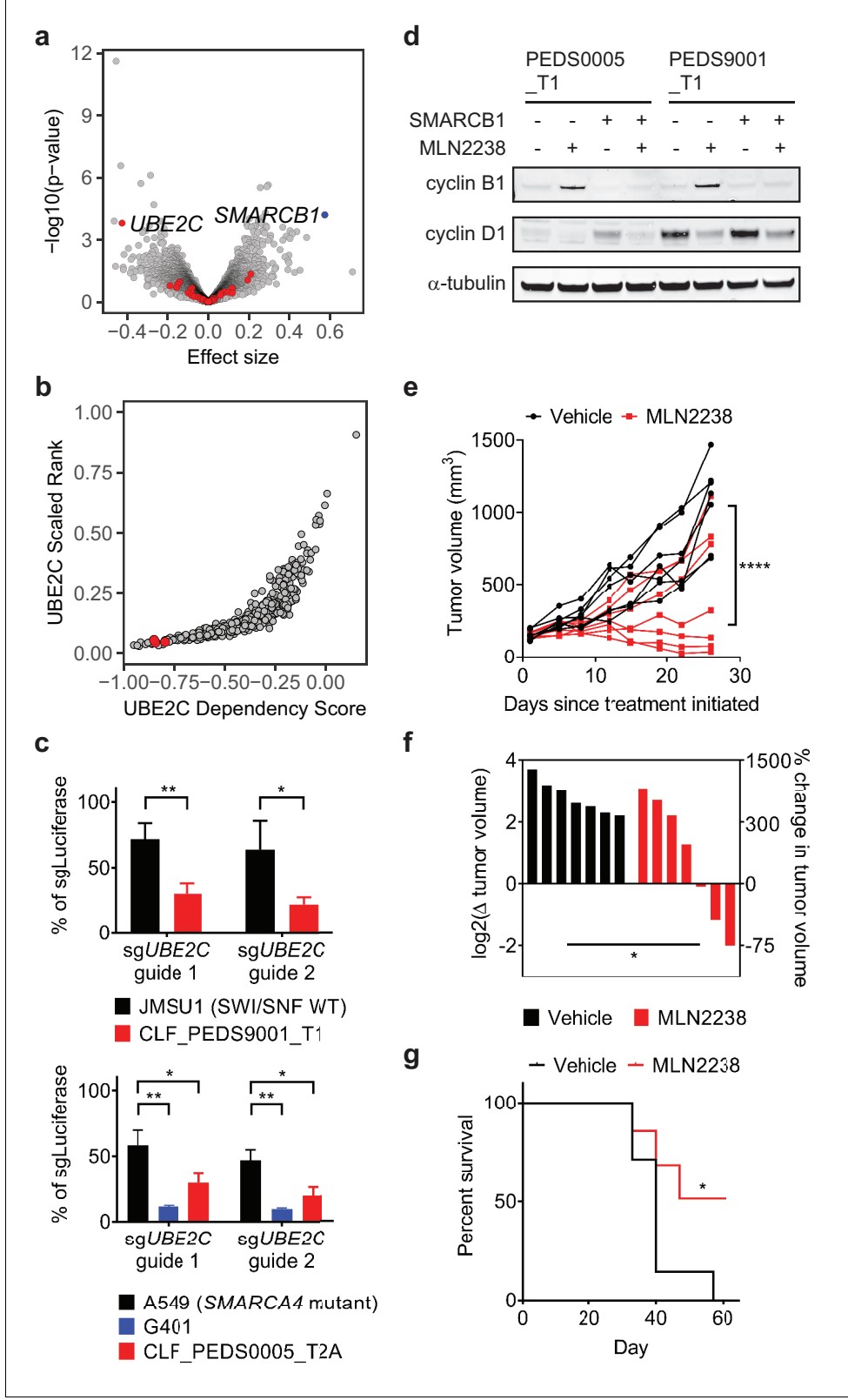

**Figure 5.** Cell cycle arrest secondary to proteasome inhibition acts via UBE2C/cyclin B1 dysregulation and proteasome inhibition with MLN2238 is effective in vivo. (a) Volcano plot identifying genes that are required for survival in SMARCB1 deficient cancers. 204 genes were significantly upregulated when comparing the log2 fold change between SMARCB1 deficient cells and SMARCB1 re-expressed cells (*Supplementary file 6*). We assessed

*Figure 5 continued on next page*

*Figure 5 continued*

how loss of these genes affected viability in 3 cell lines with loss of SMARCB1 as compared to the rest of the 482 cell lines in Project Achilles DepMap 18Q3, a genome-wide loss of function screen using CRISPR-Cas9 and calculating an effect size (e.g. differential of the 3 cell lines to 482 cell lines). A negative effect size identifies genes when deleted are required for cells for survival and the 204 genes are identified in red. Deletion of *UBE2C* was significantly depleted. Deletion of *SMARCB1* serves as a positive control in these SMARCB1 deficient cancers as these cell lines have loss of SMARCB1. (**b**) SMARCB1 deficient lines are in the top 5% of cell lines ranked by how dependent they are on *UBE2C* based on Project Achilles DepMap 18Q3 dataset. Three ATRT cancer cell lines (red dots; CHLA266, CHLA06, COGAR359) were compared to 482 cell lines profiled in Project Achilles (CERES dataset 18Q3). (**c**) Gene deletion of *UBE2C* by CRISPR-Cas9 leads to significant viability defects in RMC and MRT cell lines as compared to either SWI/SNF wt cell line, JMSU1 (day 10), or *SMARCA4* mutant cell line, A549 (day 6). Error bars shown are standard deviations from two biological replicates. * indicates a two-tailed t-test p-value<0.05 and **<0.005. (**d**) Treatment with proteasome inhibitor, MLN2238 at 200 nM, leads to upregulation of cyclin B1 and this phenotype is rescued upon SMARCB1 re-expression in both CLF_PEDS0005_T1 and CLF_PEDS9001_T1. Cyclin D1 is included as a control to ensure that the effects of proteasome inhibition are specific to cyclin B1. Blots are representative of two biological replicates. (**e**) G401 xenograft tumor growth over time by individual mouse shows that treatment effects from MLN2238 can be seen as early as 8 days from treatment initiation as compared to vehicle control. Over 26 days, tumor volumes were significantly decreased in MLN2238 treated mice. **** indicates two-way ANOVA test with p-value<0.0001. (**f**) Waterfall plot of each tumor by log2 change in tumor volume on the left y-axis and correlative percent change in tumor volume on the right y-axis following 26 days of treatment with either vehicle (black) or MLN2238 (red). * indicates a two-tailed t-test p-value<0.05. (**g**) Kaplan-Meier curves from mice with G401 xenograft tumors treated with either vehicle or MLN2238 over 61 days. * indicates a p-value of 0.0489 by log-rank (Mantel-Cox) test.

DOI: https://doi.org/10.7554/eLife.44161.016

The following source data and figure supplements are available for figure 5:

**Source data 1.** Source data for *Figure 5a*.
DOI: https://doi.org/10.7554/eLife.44161.019
**Figure supplement 1.** Pulse dosing of proteasome inhibitors leads to suppression of the proteasome, arrests cells in G2/M and leads to programmed cell death.
DOI: https://doi.org/10.7554/eLife.44161.017
**Figure supplement 2.** Pulse treatment with MLN2238 can be rescued with SMARCB1 re-expression and in vivo studies identify MLN2238 leads to activation of cleaved caspase-3 and accumulation of cyclin B1.
DOI: https://doi.org/10.7554/eLife.44161.018

We subsequently performed in vivo studies to confirm the effect of proteasome inhibition in tumor xenografts. We used the rhabdoid tumor cell line, G401, for our in vivo studies because we noted that the primary tumor cell lines (CLF_PEDS0005_T1 and CLF_PEDS9001_T) did not form subcutaneous xenograft tumors in immunodeficient mice (*Figure 5—figure supplement 2d–e*; Materials and methods). We allowed tumors to achieve an average volume of 148 mm$^3$ and then treated mice with either vehicle or MLN2238 at the maximum tolerated dose at 7 mg/kg twice a week in the Taconic NCr-nude mouse strain. Treatment with MLN2238 over 26 days induced significant tumor stabilization or regression as compared to vehicle treated tumors as assessed by absolute tumor volume (two-way ANOVA test with p-value<0.0001; *Figure 5e–f*) and did not induce significant changes in body weight as compared to vehicle-treated tumors (two-tailed t-test p-value 0.154; *Figure 5—figure supplement 2f*). Furthermore, mice treated with MLN2238 survived significantly longer [p-value 0.0489 by log-rank (Mantel-Cox) test; *Figure 5g*].

We noted that several tumors in the treatment arm had a suboptimal response to MLN2238 (*Figure 5e–f*). We harvested tumors from two pairs of mice that either showed regression or no response to MLN2238 and assessed cleaved caspase-3 and cyclin B1 levels. We found increased cyclin B1 and cleaved caspase-3 by immunoblotting in the tumor that responded to MLN2238 but did not observe increased cyclin B1 accumulation or activation of cleaved caspase-3 in the tumor without response (*Figure 5—figure supplement 2g*) suggesting that adequate inhibition of the proteasome was not achieved in mice with a suboptimal response. Combined, these results demonstrate that MLN2238 induces a cytostatic response in SMARCB1-deficient tumors in vivo.

## Discussion

We have developed faithful patient-derived models of RMC which have been genomically validated using WGS, WES, RNA-sequencing and gene expression profiling. We have shown that these models are dependent upon the loss of SMARCB1 for survival. Re-expression of SMARCB1 in RMC leads to a significant decrease in cell counts and a senescence phenotype. Biochemically, re-expression of SMARCB1 in RMC leads to stabilization of the SWI/SNF complex in the same manner as re-expression of SMARCB1 in MRT. Diagnostically, patients with RMC are often misdiagnosed with renal cell carcinoma (RCC) due to the rarity of RMC, the lack of access to SMARCB1 histological stains and unknown sickle cell status (*Beckermann et al., 2017*). Although SMARCB1 is currently included in targeted sequencing efforts nationwide (*AACR Project GENIE Consortium, 2017*), our studies along with prior studies (*Calderaro et al., 2016*; *Carlo et al., 2017*) suggest that conventional target exome sequencing may fail to identify patients with RMC.

Patients with RMC and other SMARCB1 deficient cancers have a poor prognosis despite aggressive multi-modal therapy. Using genetic and pharmacologic screens in these RMC models, we identified the ubiquitin-proteasome system as a specific vulnerability in RMC. When we looked more broadly at other SMARCB1 deficient cancers such as MRT and ATRT, we found that these models were similarly sensitive to inhibition of the ubiquitin-proteasome system. Re-expression of SMARCB1 partially rescued the sensitivity to proteasome inhibitors in RMC and MRT models.

Prior studies have implicated MYC signaling and downstream activation of ER stress as a mechanism for sensitivity to proteasome inhibitors in Kras/Tp53 mutant pancreatic cancers with Smarcb1 deficiency (*Genovese et al., 2017*; *Moreau et al., 2016*). However, the background of mutant KRAS may be contributing to these findings as mutant HRAS or KRAS cancers are sensitive to enhanced proteotoxic stress and ER stress. Furthermore, KRAS mutant cancers depend on several proteasome components in genome scale RNAi screens (*Aguirre and Hahn, 2018*). Our studies have identified that the ubiquitin-proteasome system is a core vulnerability among a compendium of druggable targets as tested by orthogonal methods of RNA interference, CRISPR-Cas9 gene deletion or small molecule inhibition. We found that proteasome inhibition in SMARCB1-deficient cancer cell lines results in G2/M arrest due to inappropriate degradation of cyclin B1.

Although in multiple myeloma cells, tumor regression has been observed in xenografts following treatment with MLN2238 (*Chauhan et al., 2011*), we found that treatment with MLN2238 of SMARCB1 deficient xenografts led to a cytostatic response. This finding is similar to what has been observed in xenograft models of non-small cell lung cancer (14 tumor models) and colon cancer (6 tumor models) (*Chattopadhyay et al., 2015*). We note that these studies were performed in mice that tolerated MLN2238 treatments at 11–14 mg/kg (*Chauhan et al., 2011*), a dose which we were unable to achieve in the Taconic Ncr-nude mice and may have led to the observed heterogeneous tumor response to MLN2238 treatment. Nevertheless, these studies still support the importance of testing this hypothesis in patients, particularly since there are no standard therapies for SMARCB1-deficient cancers.

There have been case reports of one adult and two children with RMC who exhibited extraordinary responses for 2–7 years following diagnosis after empiric therapy with bortezomib either as monotherapy or in combination with chemotherapy (*Carden et al., 2017*; *Ronnen et al., 2006*). Our findings suggest that testing oral proteasome inhibitors such as MLN2238 for patients with RMC and potentially more broadly across SMARCB1-deficient cancers is warranted.

## Materials and methods

**Key resources table**

| Reagent type (species) or resource | Designation | Source or reference | Identifiers | Additional information |
|---|---|---|---|---|
| Gene (*Homo sapiens*) | *SMARCB1* | NA | | |
| Gene (*Homo sapiens*) | *PSMB5* | NA | | |
| Gene (*Homo sapiens*) | *CCNB1* | NA | | |
| *Continued on next page* | | | | |

*Continued*

| Reagent type (species) or resource | Designation | Source or reference | Identifiers | Additional information |
|---|---|---|---|---|
| Gene (*Homo sapiens*) | *UBE2C* | NA | | |
| Strain, strain background (*Mus musculus*) | CrTac:NCr-*Foxn1*$^{nu}$ | Taconic Biosciences | RRID:IMSR_TAC:ncrnu | |
| Genetic reagent () | Luciferase | this paper | RRID:Addgene_117072 | Backbone: pXPR_BRD003 |
| Genetic reagent (*Homo sapiens*) | *UBE2C* guide 1 | this paper | RRID:Addgene_117068 | Backbone: pXPR_BRD003 |
| Genetic reagent (*Homo sapiens*) | *UBE2C* guide 2 | this paper | RRID:Addgene_117071 | Backbone: pXPR_BRD003 |
| Genetic reagent (*Homo sapiens*) | *PSMB5* guide 1 | this paper | RRID:Addgene_117073 | Backbone: pXPR_BRD003 |
| Genetic reagent (*Homo sapiens*) | *PSMB5* guide 2 | this paper | RRID:Addgene_117074 | Backbone: pXPR_BRD003 |
| Genetic reagent (*Homo sapiens*) | pDONR223 *SMARCB1* | this paper | RRID:Addgene_111181 | |
| Genetic reagent (*Homo sapiens*) | pLXI_401 *LacZ* | this paper | RRID:Addgene_111183 | Backbone: pLXI_403 |
| Genetic reagent (*Homo sapiens*) | pLXI_403 *LacZ* | this paper | RRID:Addgene_111184 | Backbone: pLXI_403 |
| Genetic reagent (*Homo sapiens*) | pLXI_401 *SMARCB1* | this paper | RRID:Addgene_111182 | Backbone: pLXI_401 |
| Genetic reagent (*Homo sapiens*) | pLXI_403 *SMARCB1* | this paper | RRID:Addgene_111185 | Backbone: pLXI_403 |
| Genetic reagent (*Homo sapiens*) | CP1050 | PMID: 27329820 | | Druggable Cancer Targets v1.0 library (shRNA) |
| Genetic reagent (*Homo sapiens*) | CP1074 | PMID: 27329820 | | Druggable Cancer Targets v1.0 library (CRISPR-Cas9) |
| Cell line (*Homo sapiens*) | CLF_PEDS0005_N | this paper | | normal kidney cell line |
| Cell line (*Homo sapiens*) | CLF_PEDS0005_T1 | this paper | | primary RMC cell line |
| Cell line (*Homo sapiens*) | CLF_PEDS0005_T2A | this paper | | metastatic RMC cell line (adherent) |
| Cell line (*Homo sapiens*) | CLF_PEDS0005_T2B | this paper | | metastatic RMC cell line (suspension) |
| Cell line (*Homo sapiens*) | CLF_PEDS9001_T1 | this paper | | primary RMC cell line |
| Cell line (*Homo sapiens*) | CLF_PEDS1012_T1 | this paper | | Wilms tumor cell line |
| Cell line (*Homo sapiens*) | G-401 | ATCC | CRL-1441 RRID:CVCL_0270 | |
| Cell line (*Homo sapiens*) | RPMI 8226 | ATCC | CCL-155 RRID:CVCL_0014 | |
| Cell line (*Homo sapiens*) | NCI-H2172 | ATCC | CRL-5930 RRID:CVCL_1537 | |
| Cell line (*Homo sapiens*) | A549 | ATCC | CCL-185 RRID:CVCL_0023 | |
| Cell line (*Homo sapiens*) | HA1E | other | | Cell line maintained in W. C. Hahn lab |
| Cell line (*Homo sapiens*) | JMSU1 | PMID: 30777879 | RRID:CVCL_2081 | |
| Antibody | anti-cytokeratin CAM5.2 | BD Biosciences | 349205 RRID:AB_2134314 | Immunohistochemistry |

*Continued on next page*

Continued

| Reagent type (species) or resource | Designation | Source or reference | Identifiers | Additional information |
|---|---|---|---|---|
| Antibody | anti-SMARCB1/BAF47 | BD Biosciences | 612110 RRID:AB_399481 | Immunohistochemistry |
| Antibody | anti-ARID1A (mouse monoclonal) | Santa Cruz Biotechnology | sc-373784 RRID:AB_10917727 | 1:500 |
| Antibody | antii-α-tubulin (mouse monoclonal) | Sigma-Aldrich | T9026 RRID:AB_477593 | 1:5000 |
| Antibody | anti-β-actin (C-4) (mouse monoclonal) | Santa Cruz Biotechnology | sc-47778 RRID:AB_2714189 | 1:10,000 |
| Antibody | anti-β-actin (D6A8) (rabbit monoclonal) | Cell Signaling Technology | #8457 RRID:AB_10950489 | 1:10,000 |
| Antibody | anti-BAF57/SMARCE1 (rabbit polyclonal) | Bethyl Laboratories, Inc. | A300-810A RRID:AB_577243 | 1:5000 |
| Antibody | anti-BAF60a (mouse monoclonal) | Santa Cruz Biotechnology | sc-135843 RRID:AB_2192137 | 1:500 |
| Antibody | anti-SMARCC1/BAF155 (D7F8S) (rabbit monoclonal) | Cell Signaling Technology | #11956 | 1:1000 |
| Antibody | anti-BAF170 (G-12) (mouse monoclonal) | Santa Cruz Biotechnology | sc-166237 RRID:AB_2192013 | 1:500 |
| Antibody | anti-SMARCA4 (G-7) (mouse monoclonal) | Santa Cruz Biotechnology | sc-17796 RRID:AB_626762 | 1:1000 |
| Antibody | anti-Cleaved Caspase-3 (Asp175) (5A1E) (rabbit monoclonal) | Cell Signaling Technology | #9664 RRID:AB_2070042 | 1:1000 |
| Antibody | anti-c-Myc (N-262) (rabbit polyclonal) | Santa Cruz Biotechnology | sc-764 RRID:AB_631276 | 1:200 |
| Antibody | anti-c-Myc (rabbit polyclonal) | Cell Signaling Technology | #9402 RRID:AB_2151827 | 1:1000 |
| Antibody | anti-cyclin B1 (V152) (mouse monoclonal) | Cell Signaling Technology | #4135 RRID:AB_2233956 | 1:1000 |
| Antibody | anti-cyclin B1 (rabbit polyclonal) | Cell Signaling Technology | #4138 RRID:AB_2072132 | 1:1000 |
| Antibody | anti-cyclin D1 (M-20) (rabbit polyclonal) | Santa Cruz Biotechnology | sc-718 RRID:AB_2070436 | 1:1000 |
| Antibody | anti-GAPDH (14C10) (rabbit monoclonal) | Cell Signaling Technology | #2118 RRID:AB_561053 | 1:1000 |
| Antibody | anti-GAPDH (D4C6R) (mouse monoclonal) | Cell Signaling Technology | #97166 RRID:AB_2756824 | 1:1000 |
| Antibody | anti-GRP78 (mouse monoclonal) | Rockland Immunochemicals | 200–301 F37 RRID:AB_2611159 | 1:10,000 |
| Antibody | anti-IRE1α (14C10) (rabbit monoclonal) | Cell Signaling Technology | #3294 RRID:AB_823545 | 1:2000 |
| Antibody | anti-Lamin A/C (rabbit polyclonal) | Cell Signaling Technology | #2032 RRID:AB_2136278 | 1:1000 |
| Antibody | anti-PSMB5 (rabbit polyclonal) | Abcam | ab3330 RRID:AB_303709 | 1:1000 |
| Antibody | anti-SMARCB1/SNF5 (rabbit polyclonal) | Bethyl Laboratories, Inc. | A301-087A RRID:AB_2191714 | 1:5000 |
| Antibody | anti-UBE2C (mouse monoclonal) | Proteintech | 66087–1 RRID:AB_11232220 | 1:1000 |
| Commercial assay or kit | QIAamp DNA Blood Midi Kit | Qiagen | 51183 | |
| Commercial assay or kit | QIAprep Spin Miniprep Kit | Qiagen | 27106 | |

*Continued*

| Reagent type (species) or resource | Designation | Source or reference | Identifiers | Additional information |
|---|---|---|---|---|
| Commercial assay or kit | Rneasy Plus Mini Kit | Qiagen | 74134 | |
| Commercial assay or kit | Qubit RNA HS Assay Kit | Thermo Fisher Scientific | Q32852 | |
| Commercial assay or kit | KAPA Stranded mRNA-Seq Kit | Kapa Biosystems | KK8420 | |
| Commercial assay or kit | KAPA Library Quantification Kit | Kapa Biosystems | KK4835 | |
| Commercial assay or kit | High-Capacity cDNA Reverse Transcription Kit | Thermo Fisher Scientific | 4368814 | |
| Commercial assay or kit | Power SYBR Green PCR Master Mix | Thermo Fisher Scientific | 4368708 | |
| Commercial assay or kit | Senescence β-Galactosidase Staining Kit | Cell Signaling Technology | 9860S | |
| Commercial assay or kit | CellTiter-Glo Luminescent Cell Viability Assay | Promega | G7570 | |
| Commercial assay or kit | Proteasome-Glo Chymotrypsin-Like Cell-based Assay | Promega | G8660 | |
| Commercial assay or kit | Annexin V: FITC Apoptosis Detection Kit | Thermo Fisher Scientific | BD 556547 | |
| Commercial assay or kit | PI/Rnase Staining Buffer | BD Pharmingen | 550825 | |
| Commercial assay or kit | FxCycle PI/Rnase Staining Solution | Invitrogen | F10797 | |
| Chemical compound, drug | bortezomib (PS-341) | Selleck Chemicals | S1013 | in vitro studies |
| Chemical compound, drug | ixazomib (MLN2238) | Selleck Chemicals | S2180 | in vitro studies |
| Chemical compound, drug | nocodazole | Selleck Chemicals | S2775 | in vitro studies |
| Chemical compound, drug | Ro-3306 | Selleck Chemicals | S7747 | in vitro studies |
| Chemical compound, drug | MLN2238 | MedChem Express | HY-10453 | in vivo studies |
| Software, algorithm | SvABA v0.2.1 | PMID: 29535149 | | |
| Software, algorithm | FlowJo v10.0 | | RRID:SCR_008520 | |
| Software, algorithm | ComBat | | RRID:SCR_010974 | |
| Software, algorithm | GraphPad Prism v8.0 | | RRID:SCR_002798 | |
| Software, algorithm | GSEA | Broad Institute | RRID:SCR_003199 | |
| Software, algorithm | GATK v.4.0.4.0 | PMID: 20644199 | RRID:SCR_001876 | |
| Software, algorithm | TopHat v2.0.11 | PMID: 22383036 | RRID:SCR_013035 | |
| Software, algorithm | DESeq2 | PMID: 25516281 | RRID:SCR_015687 | |

## Derivation of RMC models

Patients assented or families consented to IRB approved protocols. Patient PEDS0005 whole blood, adjacent normal kidney and tumor tissue were obtained within 6 hr as part of the nephrectomy following neoadjuvant chemotherapy. Upon relapse, pleural fluid was obtained from a palliative thoracentesis. The tumor and adjacent normal kidney tissue was minced into 2–3 mm$^3$ cubes. CLF_PEDS0005 primary tissue was then dissociated as previously described and cultured in both RPMI media containing 10% FBS (Sigma) or DMEM/F-12 media containing ROCK inhibitor, Y-27632, insulin, cholera toxin, 5% FBS, and penicillin/streptomycin (*Liu et al., 2012*). Samples were gently passaged when cultures achieved 80–90% confluence. The normal kidney cell line was named

CLF_PEDS0005_N. The tumor kidney cell line was named CLF_PEDS0005_T1. For the pleural fluid sample, samples were grown initially in conditioned media as previously published (*Liu et al., 2012*). Adherent and suspension cells were continuously passaged when cells reached confluence. By passage 5, cells were noted to be growing as adherent and suspension cells, and these were sub-cultured to yield CLF_PEDS0005_T2A and CLF_PEDS0005_T2B, respectively. Samples were then transitioned to DMEM/F-12 media or RPMI as above at passage 13.

Patient PEDS9001 whole blood, adjacent normal kidney and tumor tissue were obtained similarly to PEDS0005. Discarded tissue from the nephrectomy following neoadjuvant chemotherapy was sent to our institution within 24 hr of resection. For the normal kidney and tumor tissue, samples were minced to 2–3 mm$^3$ and plated onto six well plates (Corning, NY). Following mincing, tumor samples were cultured without further digestion. Tumor samples were grown in culture in the DMEM/F-12 media as described above to yield CLF_PEDS9001_T1, while the normal samples yielded a cell culture that matched the tumor cell line.

## Immunohistochemistry

All immunohistochemical staining was done in the clinical histopathology laboratory at Boston Children's Hospital with appropriate positive controls performed with each run. Antibodies used included anti-cytokeratin CAM5.2 (BD Biosciences, 349205) and SMARCB1/BAF47 (BD Biosciences, 612110).

## Breakapart fluorescent in situ hybridization (FISH)

We developed a custom dual-color breakapart FISH probe, using BAC probes surrounding SMARCB1 at 22q11.23: RP11-662F7 (telomeric to SMARCB1, labeled in green) and RP11-1112A23 (centromeric to SMARCB1, labeled in red) (Empire Genomics, Buffalo, NY). The probe set was hybridized to a normal control to confirm chromosomal locations and to determine the frequency of expected fusion signals in normal cells. 50 nuclei were scored by two independent observers (n = 100 per cell line) in the CLF_PEDS0005 and CLF_PEDS9001 models.

## Whole Exome Sequencing

We performed whole exome sequencing (WES) from genomic DNA extracted from whole blood, normal/tumor tissues, and our patient-derived cell lines as noted in the text. One microgram of gDNA (as measured on a Nanodrop 1000 (Thermo Fisher Scientific)) was used to perform standard (~60 x mean target coverage for normal) or deep (~150 x mean target coverage for tumor and cell lines) WES. Illumina (Dedham, MA) chemistry used.

## Whole Genome Sequencing

We performed PCR-free whole genome sequencing (WGS) from gDNA extracted from whole blood, normal/tumor tissues, and our patient-derived cell lines as noted in the text. Two micrograms of gDNA were used to perform standard (for normal) or deep (for tumor and cell lines) coverage. Illumina (Dedham, MA) HiSeq X Ten v2 chemistry was used. We achieved an average depth of coverage of 38x for the germline DNA and 69x for the tumor cell line DNA.

## RNA-sequencing

For *Figure 2a*, samples were processed using Illumina TruSeq strand specific sequencing. We performed poly-A selection of mRNA transcripts and obtained a sequencing depth of at least 50 million aligned reads per sample. For SMARCB1 re-expression RNAseq experiments and MLN2238 vs DMSO-treated experiments, samples were collected as biological replicates or triplicates. RNA was extracted using Qiagen RNeasy Plus Mini Kit (Qiagen, Hilden, Germany). RNA was normalized using the Qubit RNA HS Assay (Thermo Fisher Scientific). Five hundred ng of normalized RNA was subsequently used to create libraries with the Kapa Stranded mRNA-seq kit (Kapa Biosystems, KK8420; Wilmington, MA). cDNA libraries were then quantitatively and qualitatively assessed on a BioAnalyzer 2100 (Agilent, Santa Clara, CA) and by qRT-PCR with Kapa Library Quantification Kit. Libraries were subsequently loaded on an Illumina HiSeq 2500 and achieved an average read depth of 10 million reads per replicate.

## Genomic analyses

WGS - Samples were aligned to Hg19. Structural variation and indel Analysis By Assembly (SvABA) v0.2.1 was used to identify large deletions and structural variations. (*Wala et al., 2018*). WES – Samples were aligned to Hg19. Samples were analyzed using GATK v4.0.4.0 for copy number variation (CNV), single nucleotide polymorphism (SNP) and indel identification across our RMC samples simultaneously using filtering parameters set by GATK (Broad Institute, Cambridge, MA) (*McKenna et al., 2010*). MuTect 2.0 was used to identify candidate somatic mutations and these were filtered based on the Catalogue of Somatic Mutations in Cancer (COSMIC) (*Forbes et al., 2015*). RNA – CLF_PEDS0005 and CLF_PEDS9001 samples and TARGET Wilms and Rhabdoid tumor samples (dbGaP phs000218.v19.p7) were aligned or re-aligned with STAR and transcript quantification performed with RSEM. The TARGET initiative is managed by the NCI and information can be found at https://ocg.cancer.gov/programs/target. These normalized samples were then analyzed with t-SNE (*Maaten, 2014*). The following parameters were used in the t-SNE analyses: perplexity 10, theta 0, iterations 3000. RNA sequencing samples in the SMARCB1 re-expression studies were subsequently aligned and analyzed with the Tuxedo suite (e.g. aligned with TopHat 2.0.11, abundance estimation with CuffLinks, differential analysis with CuffDiff and CummeRbund) (*Trapnell et al., 2010*). For the comparison to previously published work (*Wang et al., 2017*), the published RNA sequencing samples along with our samples were re-aligned with TopHat 2.0.11 and analyzed with the Tuxedo suite. For samples treated with DMSO or MLN2238, samples were aligned as above and analyzed with DESeq2 (*Love et al., 2014*).

## Sanger sequencing confirmation of WGS findings

gDNA was extracted using QIAamp DNA mini kit (Qiagen). We performed a mixing study of our RMC cell lines with gDNA isolated from the G401 MRT cell line and then performed PCR amplification. We determined that the lower limits of detection of these fusions with our methods were ~1% of tumor cell line gDNA with a minimum 50 ng of gDNA. We subsequently performed the same PCR reactions with 100 ng of gDNA from the tumor tissue samples. Samples were gel purified and submitted for Sanger sequencing (Eton Bio). We found that the sequences from the tumor tissue samples matched those of the cell lines, confirming that the genomic alterations that we found in the cell lines reflect those found in the original tumor. Primers utilized were CLF_PEDS0005 *chr1* forward (ATAAGACATAACTTGGCCGG), CLF_PEDS0005 *SMARCB1* reverse (TTTTCCAAAAGGTTTA-CAAGGC), CLF_PEDS9001 *chr12* forward (AAAAGCATATGTATCCCTTGCT), CLF_PEDS9001 *SMARCB1* reverse (CCTCCAGAGCCAGCAGA).

## Quantitative RT-PCR

RNA was extracted as above and normalized using the Nanodrop to one microgram. One microgram of RNA was then added to the High Capacity cDNA Reverse Transcription Kit (Thermo Fisher Scientific) and PCR reactions were performed as per manufacturer's recommendations. cDNA was then diluted and added to primers (*Supplementary file 11*) and Power SYBR Green PCR Master Mix (Thermo Fisher Scientific). Samples were run on a BioRad CFX384 qPCR System in a minimum of technical quadruplicates. Results shown are representative of at least two biological replicates.

## Gene-expression array analysis

We performed Affymetrix Human Genome U133 Plus 2.0 on our RMC cell lines. We then combined the following GEO datasets using a GenePattern module with robust multi-array (RMA) normalization GSE64019, GSE70421, GSE70678, GSE36133, GSE94321 (*Barretina et al., 2012*; *Calderaro et al., 2016*; *Johann et al., 2016*; *Richer et al., 2017*; *Wang et al., 2017*). We utilized COMBAT and then tSNE to account for batch effects and to identify clusters of similarity (*Chen et al., 2011*; *Johnson et al., 2007*; *Maaten, 2014*).

## Glycerol gradients followed by SDS-PAGE

Nuclear extracts and gradients were performed as previously published (*Boulay et al., 2017*). Briefly, 500 micrograms of nuclear extract from approximately 30 million cells were resuspended in 0% glycerol HEMG buffer containing 1 mM DTT, cOmplete protease inhibitors and PhosStop (Roche). This was placed on a 10–30% glycerol gradient and ultracentrifuged at 40 k RPM for 16 hr at 4C.

Following centrifugation, fractions of 550 µL were collected. Samples were then prepared with 1x LDS Sample Buffer (Thermo). Samples were run on a 4–12% Bis-Tris gel and then transferred by immunoblotting in tris-glycine-SDS buffer with methanol. Immunoblots were subsequently blocked with Licor Blocking Buffer (Lincoln, NE) and then incubated with antibodies as noted in the methods section. Immunoblots shown are representative of at least two biological replicates.

## Cell lines

Primary cell lines were authenticated by Fluidigm or WES/WGS sequencing or by qRT-PCR. Cells were tested for mouse antibody production (Charles Rivers Laboratories; Wilmington, MA) and mycoplasma using the Lonza MycoAlertPLUS Mycoplasma Detection Kit (Morristown, NJ). Established cell lines were authenticated by Fluidigm SNP testing. Cell lines were refreshed after approximately 20 passages from the frozen stock.

## Small molecules

Bortezomib, MLN2238, nocodazole and RO-3306 were purchased from SelleckChem (Houston, TX) for the *in vitro* studies. Compounds were resuspended in DMSO and frozen down in 20 microliter aliquots to limit freeze-thaw cycles. Compounds were added as noted in the figure legends. *In vitro* studies used 15 nM for bortezomib and 100 nM for MLN2238 or are otherwise specified in the text. For the pulse experiments, we used 2.5 micromolar of MLN2238.

## SMARCB1 induction studies

pDONR223 SMARCB1 was Sanger sequenced (Eton Bio) and aligned to variant 2 of SMARCB1. SMARCB1 was subsequently cloned into the inducible vector pLXI401 or pLXI403 (Genomics Perturbation Platform at the Broad Institute, Cambridge, MA) by Gateway Cloning. LacZ was used as a control. Lentivirus was produced using tet-free serum (Clontech, Mountain View, CA). Cell lines were infected with lentivirus to generate stable cell lines. Cell lines were then confirmed to re-express LacZ or SMARCB1 by titrating levels of doxycycline (Clontech). Parental cell lines were treated with increasing doses of doxycycline to determine the toxicity to cells and measured by Cell-TiterGlo after 96 hr. Cells were then grown with or without doxycycline in a six well plate. Cells were counted by Trypan blue exclusion on a ViCELL XR (Beckman Coulter, Brea, CA) every 4–5 days. Results shown are the average of at least three biological replicates.

## Senescence assays

Cells were plated in a six well dish and treated with or without doxycycline for up to 7 days. Senescence was assessed with the Senescence β-Galactosidase Staining Kit without modifications (Cell Signaling Technologies, Danvers, MA).

## Druggable Cancer Targets (DCT) v1.0 shRNA/sgRNA libraries, pooled screens and small-molecule profiling

These were performed as previously published (*Hong et al., 2016*). Briefly, we utilized the DCT v1.0 shRNA (CP1050) and sgRNA (CP0026) libraries from the Broad Institute Genetic Perturbation Platform (GPP) (http://www.broadinstitute.org/rnai/public/). Viruses from both pools were generated as outlined at the GPP portal. As CLF_PEDS0005_T2A and CLF_PEDS0005_T2B were expanded, we performed titrations with the libraries as outlined at the GPP portal. For the sgRNA pool, both cell lines were first transduced with Cas9 expression vector pXPR_BRD111. We screened the DCT v1.0 shRNA library in biological replicates and the Cas9 expressing cell lines with the sgRNA pools at an early passage (<20) and at a multiplicity of infection (MOI) <1, at a mean representation rate above 500 cells per sgRNA or shRNA. gDNA was extracted and was submitted for sequencing of the barcodes. We achieved sequencing depths of at least 500 reads per shRNA or sgRNA.

## CRISPR-Cas9 validation studies

sgRNAs targeting the genes noted in the manuscript (e.g. PSMB5, UBE2C and controls; *Supplementary file 12*) were generated and introduced into the pXPR_BRD003 backbone. These were then sequence confirmed by Sanger sequencing (Eton Biosciences). Lentivirus was produced

and used for infection to generate stable cell lines expressing Cas9. Cells were counted or harvested for protein as noted in the text.

## Immunoblots

After indicated treatments, cell lysates were harvested using RIPA buffer (Cell Signaling Technologies) with protease inhibitors (cOmplete, Roche) and phosphatase inhibitors (PhosSTOP, Roche). Antibodies used were as follows: ARID1A (Santa Cruz; sc-373784), α-tubulin (Santa Cruz; sc-5286), β-Actin (C-4) (Santa Cruz; sc-47778), β-Actin (Cell Signaling; 8457), BAF57/SMARCE1 (Bethyl Laboratories, A300-810A), BAF60a (Santa Cruz; sc-135843), BAF155 (Cell Signaling; 11956), BAF170 (Santa Cruz; sc-166237), SMARCA4 (Santa Cruz; 17798), Cleaved Caspase-3 (Cell Signaling; 9664), c-MYC (Santa Cruz; sc-764) or c-MYC (Cell Signaling; 9402), cyclin B1 (Cell Signaling; 4135 and 4138), cyclin D1 (Santa Cruz; sc-718), GAPDH (Cell Signaling; 2118S and 97166S), GRP78 (Rockland Antibodies, Limerick, PA; 200–301 F36), IRE1-alpha (Cell Signaling; 3294), lamin A/C (Cell Signaling; 2032), PSMB5 (Abcam, Cambridge, MA; ab3330), SMARCB1/SNF5 (Bethyl A301-087A), UBE2C (Proteintech, Rosemont, IL; 66087–1). Results shown are representative of at least two biological replicates.

## Cell cycle and Annexin V/PI

Cells were treated using the conditions noted in the text. One million cells were spun down and resuspended in PBS. Cells were then subjected to FITC Annexin V and PI staining as described (BD Pharmigen; 556547). Another set of cells were subjected to PI/RNAse staining (BD Pharmingen; 550825 or Invitrogen F10797). Samples were analyzed within 1 hr with the SA3800 Spectral Analyzer (Sony Biotechnology). Biological replicates were performed. Data were analyzed with FlowJo v10 (FlowJo, Ashland, OR).

## Proteasome function assay

We measured the cell's ability to cleave Suc-LLVY-aminoluciferin utilizing Proteasome-Glo (Promega) following a one-hour treatment with the noted proteasome inhibitor and measured luminescence. Results shown are from at least two biological replicates.

## In vivo tumor injections

This research project has been reviewed and approved by the Dana-Farber Cancer Institute's Animal Care and Use Committee (IACUC), in compliance with the Animal Welfare Act and the Office of Laboratory Welfare (OLAW) of the National Institutes of Health (NIH). Five million cells of G401 in 100 µL of a 50% PBS/50% Matrigel (BD Biosciences) mixture were injected subcutaneously into flanks unilaterally in Taconic NCr-Nude (CrTac:NCr-Foxn1nu) female mice at 7 weeks of age. When tumors reached approximately 150 mm$^3$, mice were randomized into various treatment groups: vehicle control (5% 2-hydroxypropyl-beta-cyclodextrin (HPbCD)) or MLN2238 (7 mg/kg IV twice a week for 4 weeks). MLN2238 (diluted in 5% 2-HPbCD) was purchased from MedChem Express. Randomizations to the treatment arm occurred. Blinding was not performed. Statistical analysis was performed using the two-tailed t-test or Mantel-Cox as noted in the text.

## Acknowledgments

We thank our patients, the patient advocacy foundations and the RMC Alliance. We thank John Daley at the Dana-Farber Cancer Institute Flow Cytometry Core Facility, Anita Hawkins at the Brigham and Women's Hospital CytoGenomics Core, Zach Herbert at the Molecular Biology Core Facilities, Tamara Mason at the Genomic Platform at the Broad Institute, and the Boston University Microarray Resource. We thank the Hahn lab, Boehm lab, Cichowski lab, Kim Stegmaier, Rosalind Segal, David Kwiatkowski, Seth Alper, Toni Choueiri, Carlos Rodriguez-Galindo, the COG Renal Tumors Committee for critical discussions and/or reading of the manuscript. This work was supported in part by the NCI Cancer Target Discovery and Development Network U01 CA176058 (WCH) and U01 CA217848 (SLS), Katie Moore Foundation (JSB), Merkin Family Foundation (JSB), T32 GM007753 (TPH), T32 GM007226 (TPH), AACR-Conquer Cancer Foundation of ASCO Young Investigator Translational Cancer Research Award (ALH), CureSearch for Children's Cancer (ALH), NCI T32 CA136432-09 (ALH), NICHD K12 HD052896-08 (ALH), Pedals for Pediatrics (ALH), Friends

for Dana Farber (ALH), Alex's Lemonade Stand Young Investigator Grant (ALH), Boston Children's Hospital OFD BTREC CDA (ALH), NCI P50CA101942 (ALH), Cure AT/RT (SNC/ALH), Team Path to Cure (ALH), American Cancer Society MRSG-18-202-01-TBG (ALH), the Wong Family Award (ALH) and Department of Defense Prostate Cancer Research Program Postdoctoral Training Award W81XWH-15-1-0659 (GJS).

## Additional information

### Competing interests

Pratiti Bandopadhayay, Rameen Beroukhim: is a consultant for Novartis (Cambridge, MA). Paul A Clemons: is an adviser for Pfizer, Inc (Groton, CT). Cigall Kadoch: is a Scientific Founder, member of the Board of Director, Scientific Advisory Board member, Shareholder, and Consultant for Foghorn Therapeutics, Inc (Cambridge, MA). Disclosure information for CK is also found at: http://www.kadochlab.org. William C Hahn: is a consultant for Thermo Fisher, Aju IB, MPM Capital and Paraxel. W.C. H. is a founder and shareholder and serves on the scientific advisory board of KSQ Therapeutics. The other authors declare that no competing interests exist.

### Funding

| Funder | Grant reference number | Author |
| --- | --- | --- |
| Alex's Lemonade Stand Foundation for Childhood Cancer | Young Investigator Award | Andrew L Hong |
| American Association for Cancer Research | 14-40-31-HONG | Andrew L Hong |
| American Cancer Society | 132943-MRSG-18-202-01-TBG | Andrew L Hong |
| Boston Children's Hospital | OFD BTREC CDA | Andrew L Hong |
| Cure AT/RT | | Andrew L Hong<br>Susan N Chi |
| CureSearch for Children's Cancer | 328545 | Andrew L Hong |
| Dana-Farber Cancer Institute | Wong Family Award | Andrew L Hong |
| Eunice Kennedy Shriver National Institute of Child Health and Human Development | K12 HD052896 | Andrew L Hong |
| Katie Moore Foundation | | Jesse S Boehm |
| Merkin Family Foundation | | Jesse S Boehm |
| National Cancer Institute | U01 CA176058 | William C Hahn |
| National Cancer Institute | U01 CA217848 | Stuart L Schreiber |
| National Cancer Institute | P50 CA101942 | Andrew L Hong |
| National Institute of General Medical Sciences | T32 GM007753 | Thomas P Howard |
| National Institute of General Medical Sciences | T32 GM007226 | Thomas P Howard |
| Team Path to Cure | | Andrew L Hong |
| U.S. Department of Defense | W81XWH-15-1-0659 | Gabriel J Sandoval |

The funders had no role in study design, data collection and interpretation, or the decision to submit the work for publication.

## Author contributions

Andrew L Hong, Conceptualization, Data curation, Software, Formal analysis, Supervision, Funding acquisition, Validation, Investigation, Visualization, Methodology, Writing—original draft, Project administration, Writing—review and editing; Yuen-Yi Tseng, Resources, Data curation, Supervision, Investigation, Methodology; Jeremiah A Wala, Resources, Software, Formal analysis, Visualization, Methodology, Writing—review and editing; Won-Jun Kim, Data curation, Validation, Investigation, Methodology, Writing—review and editing; Bryan D Kynnap, Data curation, Investigation, Methodology; Mihir B Doshi, Data curation, Validation, Investigation, Visualization; Guillaume Kugener, Data curation, Software, Formal analysis, Visualization, Methodology; Gabriel J Sandoval, Investigation, Methodology, Writing—review and editing; Thomas P Howard, Investigation, Visualization, Methodology, Writing—review and editing; Ji Li, Validation, Investigation, Writing—review and editing; Xiaoping Yang, Michelle Tillgren, Investigation, Methodology, Project administration; Mahmhoud Ghandi, Resources, Software, Supervision, Methodology; Abeer Sayeed, Rebecca Deasy, Katherine M Labella, Validation, Investigation, Methodology; Abigail Ward, Resources, Investigation, Project administration; Brian McSteen, Barbara Van Hare, David Sandak, Resources, Funding acquisition, Project administration; Paula Keskula, Resources, Validation, Methodology, Project administration; Adam Tracy, Cora Connor, Resources, Validation, Project administration; Catherine M Clinton, Resources, Supervision, Methodology, Project administration; Alanna J Church, Resources, Supervision, Investigation, Methodology; Brian D Crompton, Katherine A Janeway, Resources, Supervision, Project administration; Ole Gjoerup, Supervision, Methodology, Project administration, Writing—review and editing; Pratiti Bandopadhayay, Software, Formal analysis, Supervision, Project administration; Paul A Clemons, Resources, Software, Supervision, Writing—review and editing; Stuart L Schreiber, David E Root, Resources, Supervision, Funding acquisition, Project administration, Writing—review and editing; Prafulla C Gokhale, Resources, Supervision, Methodology, Project administration, Writing—review and editing; Susan N Chi, Supervision, Funding acquisition, Project administration, Writing—review and editing; Elizabeth A Mullen, Resources, Supervision, Project administration, Writing—review and editing; Charles WM Roberts, Resources, Supervision, Funding acquisition, Methodology, Project administration; Cigall Kadoch, Rameen Beroukhim, Keith L Ligon, Jesse S Boehm, Resources, Supervision, Funding acquisition, Methodology, Project administration, Writing—review and editing; William C Hahn, Conceptualization, Resources, Supervision, Funding acquisition, Methodology, Writing—original draft, Project administration, Writing—review and editing

## Author ORCIDs

Andrew L Hong (iD) http://orcid.org/0000-0003-0374-1667
Rameen Beroukhim (iD) http://orcid.org/0000-0001-6303-3609
Jesse S Boehm (iD) http://orcid.org/0000-0002-6795-6336
William C Hahn (iD) http://orcid.org/0000-0003-2840-9791

## Ethics

Human subjects: Patients assented and / or families consented to Dana-Farber Cancer Institute IRB approved protocols: 11-104, 16-031.
Animal experimentation: This research protocol (04-111) has been reviewed and approved by the Dana-Farber Cancer Institute's Animal Care and Use Committee (IACUC), in compliance with the Animal Welfare Act and the Office of Laboratory Welfare (OLAW) of the National Institutes of Health (NIH).

## Decision letter and Author response

Decision letter https://doi.org/10.7554/eLife.44161.050
Author response https://doi.org/10.7554/eLife.44161.051

# Additional files

## Supplementary files

• Supplementary file 1. Significant mutations identified by MuTect2.

DOI: https://doi.org/10.7554/eLife.44161.020

• Supplementary file 2. SMARCB1 Fluorescence In Situ Hybridization results.
DOI: https://doi.org/10.7554/eLife.44161.021

• Supplementary file 3. Structural changes identified by SvABA in CLF_PEDS0005_T.
DOI: https://doi.org/10.7554/eLife.44161.022

• Supplementary file 4. Structural changes identified by SvABA in CLF_PEDS9001_T.
DOI: https://doi.org/10.7554/eLife.44161.023

• Supplementary file 5. Fusion sequences identified by PCR-Free Whole Genome Sequencing.
DOI: https://doi.org/10.7554/eLife.44161.024

• Supplementary file 6. Average differential expression across inducible SMARCB1 RMC and MRT cell lines following SMARCB1 re-expression along with comparison to TARGET.
DOI: https://doi.org/10.7554/eLife.44161.025

• Supplementary file 7. Overlap between RNAi, CRISPR-Cas9 and small-molecule screens.
DOI: https://doi.org/10.7554/eLife.44161.026

• Supplementary file 8. Gene Ontology Gene Set Enrichment Analysis from SMARCB1 re-expression studies.
DOI: https://doi.org/10.7554/eLife.44161.027

• Supplementary file 9. Average differential expression across SMARCB1 RMC and MRT cell lines following DMSO or MLN2238 treatment.
DOI: https://doi.org/10.7554/eLife.44161.028

• Supplementary file 10. Gene Ontology Gene Set Enrichment Analysis from cells treated with MLN2238.
DOI: https://doi.org/10.7554/eLife.44161.029

• Supplementary file 11. SMARCB1 exon-exon junction qRT-PCR primers.
DOI: https://doi.org/10.7554/eLife.44161.030

• Supplementary file 12. sgRNAs used in the CRISPR-Cas9 validation studies.
DOI: https://doi.org/10.7554/eLife.44161.031

• Transparent reporting form
DOI: https://doi.org/10.7554/eLife.44161.032

## Data availability

Data and materials availability: Noted plasmids in the text are available through Addgene or the Genomics Perturbations Platform at the Broad Institute of Harvard and MIT. CLF_PEDS0005_T1, CLF_PEDS0005_T2B, CLF_PEDS0005_T2A and CLF_PEDS9001_T1 cell lines are available through the Cancer Cell Line Factory at the Broad Institute of Harvard and MIT. Sequencing data reported in this paper (whole-genome sequencing and whole-exome sequencing) has been deposited in the database of Genotypes and Phenotypes (dbGaP) and GEO GSE111787.

The following datasets were generated:

| Author(s) | Year | Dataset title | Dataset URL | Database and Identifier |
|-----------|------|---------------|-------------|-------------------------|
| Hong AL, Tseng YY, Wala JA, Kim WJ, Kynnap BD, Doshi MB, Kugener G, Sandoval GJ, Howard TP, Li J, Yang X, Tillgren M, Ghandi M, Sayeed A, Deasy R, Ward A, McSteen B, Labella KM, Keskula P, Tracy A, Connor C, Clinton CM, Church AJ, Crompton BD, Janeway KA, Van Hare B, Sandak D, Gjoerup | 2019 | Renal medullary carcinomas depend upon SMARCB1 loss and are sensitive to proteasome inhibition | https://www.ncbi.nlm.nih.gov/geo/query/acc.cgi?acc=GSE111787 | NCBI Gene Expression Omnibus, GSE111787 |

| Author(s) | Year | Dataset title | Dataset URL | Database and Identifier |
|---|---|---|---|---|
| O, Bandopadhayay P, Clemons PA, Schreiber SL, Root DE, Gokhale PC, Chi SN | | | | |
| Andrew L Hong, Yuen-Yi Tseng, Jeremiah A Wala, Won-Jun Kim, Bryan D Kynnap, Mihir B Doshi, Guillaume Kugener, Gabriel J Sandoval, Thomas P Howard, Ji Li, Xiaoping Yang, Michelle Tillgren, Mahmhoud Ghandi, Abeer Sayeed, Rebecca Deasy | 2019 | Genomics of pediatric renal medullary carcinomas | http://www.ncbi.nlm.nih.gov/projects/gap/cgi-bin/study.cgi?study_id=phs001800.v1.p1 | NCBI dbGaP, phs001800.v1.p1 |

The following previously published datasets were used:

| Author(s) | Year | Dataset title | Dataset URL | Database and Identifier |
|---|---|---|---|---|
| National Cancer Institute | 2017 | National Cancer Institute (NCI) TARGET: Therapeutically Applicable Research to Generate Effective Treatments | https://www.ncbi.nlm.nih.gov/projects/gap/cgi-bin/study.cgi?study_id=phs000218.v19.p7 | NCBI, phs000218.v19.p7 |
| Han ZY, Richer W, Fréneaux P, Chauvin C | 2016 | Mouse Smarcb1-deficient models recapitulate subtypes of human rhabdoid tumors. | https://www.ncbi.nlm.nih.gov/geo/query/acc.cgi?acc=GSE64019 | NCBI Gene Expression Omnibus, GSE64019 |
| Calderaro J, Masliah-Planchon J, Richer W, Maillot L | 2016 | SMARCB1-deficient rhaboid tumors of the kidney and renal medullary carcinomas. | https://www.ncbi.nlm.nih.gov/geo/query/acc.cgi?acc=GSE70421 | NCBI Gene Expression Omnibus, GSE70421 |
| Johann PD, Erkek S, Zapatka M, Kerl K | 2016 | Gene expression data from ATRT tumor samples | https://www.ncbi.nlm.nih.gov/geo/query/acc.cgi?acc=GSE70678 | NCBI Gene Expression Omnibus, GSE70678 |
| Barretina J, Caponigro G, Stransky N, Venkatesan | 2012 | Expression data from the Cancer Cell Line Encyclopedia (CCLE) | https://www.ncbi.nlm.nih.gov/geo/query/acc.cgi?acc=GSE36133 | NCBI Gene Expression Omnibus, GSE36133 |
| Richer W, Masliah-Planchon J, Clement N, Jimenez I | 2017 | Embryonic signature distinguishes pediatric and adult rhabdoid tumors from other SMARCB1-deficient cancers | https://www.ncbi.nlm.nih.gov/geo/query/acc.cgi?acc=GSE94321 | NCBI Gene Expression Omnibus, GSE94321 |

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
