## [Decision Letter]

Thank you for submitting your article "Renal medullary carcinomas depend upon *SMARCB1* loss and are sensitive to proteasome inhibition" for consideration by *eLife*. Your article has been reviewed by three peer reviewers, including Ross Levine as the Reviewing Editor and Reviewer #1, and the evaluation has been overseen by Jeffrey Settleman as the Senior Editor. The following individuals involved in review of your submission have also agreed to reveal their identity: James Brugarolas (Reviewer #2) and Cory Abate-Shen (Reviewer #3).

The reviewers have discussed the reviews with one another and the Reviewing Editor has drafted this decision to help you prepare a revised submission.

The work is novel and of interest, and the reviewers found merit in the findings and the potential for advancing the field. The observation that SMARCB1 loss induces proteasome inhibitor sensitivity is of clear mechanistic and therapeutic relevance and could be translated to a mechanism-informed clinical trial.

Essential revisions:

1) The first section of the manuscript describing the genomic characterization of the RMC cell lines and samples is merely using brute force genomics to show what we already know. This should be markedly reduced in description as it is confirmatory. Moreover, the approach to developing the models needs to be discussed here. Not the methods per se but the approach – e.g. general comment about the patients, where are the tissues implanted, take rate etc.

2) What are the genes in GSEA which lead to similar gene expression in SMRCAB1 altered tumors in different cancer contexts? Does re-expression of SMRCAB1 lead to reversal of the aberrant gene expression; if so, for which genes and with what kinetic does this occur?

3) The major aspect which is of interest is the potential mechanism for increased proteasome inhibitor sensitivity, but this is not given enough emphasis. Is this really all due to differential cell cycle effects, e.g. like PLK1 which is always more dependent in more rapidly dividing cells? Can a Cyclin B1 rescue experiment be used to show this is the key target for proteasome inhibitor sensitivity?

4) The in vivo data are underwhelming with only a modest improvement in survival and slowing of tumor growth (not regression). Is this an issue with relative dependency on the target or due to incomplete target inhibition? Alternatively, the survival curves in Figure 5F could be continued to assess whether there is a meaningful difference if that is feasible in the time-frame.

5) While some pediatric tumors are characterized by deletion of both SMARCB1 alleles, RMC typically exhibits deletion of one allele and translocation of the other. The authors argue that, at least in their two cell lines, and more broadly, these translocations are simply LOF events. However, why they involve translocations is not clear. It would be helpful to integrate their two translocations with those reported in the literature and analyze for shared features.

---

## [Author Response]

Essential revisions:1) The first section of the manuscript describing the genomic characterization of the RMC cell lines and samples is merely using brute force genomics to show what we already know. This should be markedly reduced in description as it is confirmatory. Moreover, the approach to developing the models needs to be discussed here. Not the methods per se but the approach – e.g. general comment about the patients, where are the tissues implanted, take rate etc.

We agree with the reviewers and have revised the first section of the manuscript by streamlining the genomic characterization and focusing more on the approach in developing models of RMCs (subsection “Derivation and genomic characterization of RMC models” of the revised manuscript).

2) What are the genes in GSEA which lead to similar gene expression in SMRCAB1 altered tumors in different cancer contexts? Does re-expression of SMRCAB1 lead to reversal of the aberrant gene expression; if so, for which genes and with what kinetic does this occur?

To address this question, we first performed a comparison of the transcriptomes between the primary *SMARCB1*-deficient tumors with matched normal tissues using RNA sequencing data from the Therapeutically Applicable Research to Generate Effective Treatments (TARGET) database. We identified 6,311 differentially expressed genes with a q-value <0.1.

To determine the set of aberrant genes from *SMARCB1*-deficient tumors which normalized upon re-expression of SMARCB1, we compared these 6,311 genes with the 527 differentially expressed genes that we had identified when we re-expressed SMARCB1 across multiple SMARCB1-deficient cancer cell lines (Revised Supplementary file 6). We found 257 genes that overlapped between these two datasets. Among these genes, we found that the most significantly enriched gene sets involved the cell cycle using GSEA Gene Ontology (Revised Supplementary file 6).

We then analyzed CLF_PEDS9001_T at time points spanning 24-120 hours after re-expressing SMARCB1 to assess the kinetics of gene expression changes. We first performed qRT-PCR and immunoblotting to identify the time point where SMARCB1 elevated mRNA and protein expression occurs. We found that SMARCB1 mRNA and protein expression occurs as early as 24 hours but persists over the course of 120 hours (Figure 2—figure supplement 1E).

We then analyzed 5 of these 257 genes that are implicated in regulation of the G1/S (*RRM2, TOP2A*) or G2/M (*PLK1, CCNB1, UBE2C*) phases of the cell cycle. Specifically, we determined the kinetics over which changes in expression occurred upon re-expression of SMARCB1. For *PLK1* and *CCNB1*, we observed a gradual decrease over the course of 120 hours whereas *RRM2, UBE2C* and *TOP2A* exhibited a more profound decrease in expression after the first 24 hours and then a modest decrease over the following 96 hours (Figure 2—figure supplement 1F).

We have since revised our manuscript to include these data, figures and results (see subsection “Patient-derived models of RMC are similar to SMARCB1 deficient cancers”).

3) The major aspect which is of interest is the potential mechanism for increased proteasome inhibitor sensitivity, but this is not given enough emphasis. Is this really all due to differential cell cycle effects, e.g. like PLK1 which is always more dependent in more rapidly dividing cells? Can a Cyclin B1 rescue experiment be used to show this is the key target for proteasome inhibitor sensitivity?

To address this question, we first analyzed the experiment described in the comments raised in #2. The genes that were reversed upon SMARCB1 re-expression were those related to the cell cycle. Since we rescued the sensitivity to proteasome inhibitors by re-expression of SMARCB1, we speculated that treatment with proteasome inhibitors led to the observed effects on the cell cycle.

A cyclin B1 rescue experiment (e.g. either generating stable cell lines that no longer require cyclin B1 for survival or suppressing cyclin B1 following accumulation) is not feasible. Cyclin B2 expression compensates for loss of cyclin B1 in cancer cells, and double knockout leads to complete cell arrest (Bellanger et al., 2007; Li et al., 2018).

As an alternative, we asked whether treatment of SMARCB1-deficient cancers with cell cycle inhibitors led only to cell cycle arrest or arrest followed by cell death. For example, nocodazole is an antimitotic agent that interferes with microtubule polymerization, arrests cells in G2/M and leads to accumulation of cyclin B1 (Wolf et al., 2006). In addition, CDK1 and cyclin B1 form a complex required for mitotic progression, which can be inhibited by a CDK1 small molecule inhibitor, RO-3306 (Vassilev et al., 2006). We treated both G401 and CLF_PEDS9001_T with nocodazole or RO-3306 for 24 hours and observed accumulation of cells in G2/M similar to that of MLN2238 (Figure 4—figure supplement 2D). We also found increased cyclin B1 and cleaved caspase-3 following treatment with these compounds (Figure 4—figure supplement 2E). By 72 hours, we saw that treatment with either nocodazole or RO-3306 induced cell death in the majority of cells (65-90%) (Figure 4—figure supplement 2F), similar to what we observed when we treated cells with MLN2238 (Figure 4C). These observations suggest that SMARCB1-deficient cell lines are susceptible to programmed cell death following treatment with a cell cycle inhibitor and that the cell cycle arrest seen from treatment with MLN2238 can lead to programmed cell death.

Based on these studies, we conclude that proteasome inhibitors induce the G2/M cell cycle arrest phenotype in our SMARCB1 deficient cell lines. We have incorporated these new experiments, expanded our results in the revised manuscript (see subsection “Proteasome inhibition leads to cell cycle arrest in G2/M and subsequent cell death”).

*4) The* in vivo *data are underwhelming with only a modest improvement in survival and slowing of tumor growth (not regression). Is this an issue with relative dependency on the target or due to incomplete target inhibition? Alternatively, the survival curves in Figure 5F could be continued to assess whether there is a meaningful difference if that is feasible in the time-frame.*

During our tolerability and dose finding studies, we found that the maximal dose of MLN2238 tolerated by the Taconic NCR-nude mice at 6-8 weeks of age was 7 mg/kg by intravenous injection. Other investigators using multiple myeloma cells in the Taconic CB-17 mouse strain dosed the MLN2238 compound at 11 mg/kg (Chauhan et al., 2011) while in lung and colon cancer xenograft models, investigators used doses ranging from 11-14 mg/kg in other mouse strains (Chattopadhyay et al., 2015). We used the Taconic NCr-Nude mice because the G401 cells form tumors efficiently in this mouse strain. We speculate that the lower dose that we used for these experiments led to inadequate distribution of MLN2238 in a subset of tumors (see below).

We agree with the reviewers that the presentation of subpanels in our original submission made the effect of treatment look modest. Specifically, we found that treatment with MLN2238 led to tumor regression or stable disease in 4 animals but failed to affect tumor growth in 3 tumors. We included two plots in our original submission. First was Figure 5E (original submission) which represented the average of all tumors treated with vehicle or MLN2238 and second was Supplementary figure 7F (original submission) which showed a waterfall plot of response after 26 days of vehicle or MLN2238 treatment. On reflection, the waterfall plot is more representative of the response of the tumors as some tumors responded to treatment and others did not (see below). To better represent our findings, we have replaced Figure 5E (revised submission) to show the individual response of each mouse. We also moved and modified Supplementary figure 7F (original submission) to Figure 5F (revised submission) in the revised manuscript.

To assess the heterogeneous response to treatment, we performed new experiments in which we assessed whether the tumors showed evidence of response to MLN2238 by analyzing cyclin B1 and cleaved caspase-3 protein expression. We compared two tumors from our vehicle treated mice to one tumor that responded to MLN2238 and one tumor that did not respond to MLN2238. We observed increased cyclin B1 and cleaved caspase-3 by immunoblot in the tumor that responded to MLN2338 treatment but no evidence of accumulation of cyclin B1 or activation of cleaved caspase-3 in the tumor that did not respond to MLN2238 (revised submission Figure 5—figure supplement 2G). These observations suggest that for the tumors that failed to respond to treatment with MLN2238, we did not achieve the necessary inhibition of the proteasome to lead to cyclin B1 accumulation and subsequent activation of cleaved caspase-3.

Although in multiple myeloma cells, tumor regression has been observed in xenografts following treatment with MLN2238 (Chauhan et al., 2011), we found that treatment of SMARCB1 deficient xenografts led to a cytostatic response. This finding is similar to what has been observed in xenograft models of non-small cell lung cancer (14 tumor models) and colon cancer (6 tumor models) (Chattopadhyay et al., 2015). Nevertheless, these studies still support the importance of testing this hypothesis in patients, particularly since there are no standard therapies for SMARCB1-deficient cancers.

In the revised manuscript we now include these revised figures and have revised the text to explain these findings more clearly in the Results section and Discussion section.

5) While some pediatric tumors are characterized by deletion of both SMARCB1 alleles, RMC typically exhibits deletion of one allele and translocation of the other. The authors argue that, at least in their two cell lines, and more broadly, these translocations are simply LOF events. However, why they involve translocations is not clear. It would be helpful to integrate their two translocations with those reported in the literature and analyze for shared features.

There are two published articles that described the translocations seen in patients with RMC (Calderaro et al., 2016; Carlo et al., 2017). In Carlo et al., the authors used FISH to identify the fusion event in 8 of 10 patients but were unable to identify the specific translocations either due to the lack of whole genome data or the lack of adequate tissue sample. In Calderaro et al., the authors identified fusion events in 4 of 5 patients. We analyzed the available data in Calderaro et al. in the context of our genomic findings. We failed to find shared featured when we aligned the sequences (Figure 1—figure supplement 1G). We found at most 2 bp of homology on each side of the breakpoints which makes homology-mediated mechanisms unlikely. We then used RepeatMasker 4.0.8 (www.repeatmasker.org), an online database of repetitive DNA elements, to identify shared breakpoint sequences but we were unable to find any matches. As with most fusion events, the molecular basis for these RMC fusion events remains undefined.

We have summarized these findings in subsection “Derivation and genomic characterization of RMC models” of the revised manuscript.